# Expression of foetal gene *Pontin* is essential in protecting heart against pathological remodelling and cardiomyopathy

Bayu Lestari[1,4,5], Ardiansah Bayu Nugroho[1,5], Thuy Anh Bui[1,5], Binh Nguyen[1], Nicholas Stafford[1,2], Sukhpal Prehar[1], Min Zi[1], Ryan Potter [1], Efta Triastuti[1], Florence M. Baudoin[1], Alicia D'Souza[3], Xin Wang[1], Elizabeth J. Cartwright[1] & Delvac Oceandy [1] ✉

Cardiac remodelling is a key process in the development of heart failure. Reactivation of foetal cardiac genes is often associated with cardiac remodelling. Here we study the role of Pontin (*Ruvbl1*), which is highly expressed in embryonic hearts, in mediating adverse remodelling in adult mouse hearts. We observe that Pontin deficiency in cardiomyocytes leads to induced apoptosis, increased hypertrophy and fibrosis, whereas Pontin overexpression improves survival, increases proliferation and reduces the hypertrophic response. Moreover, RNAseq analysis show that genes involved in cell cycle regulation, cell proliferation and cell survival/apoptosis are differentially expressed in Pontin knockout. Specifically, we detect changes in the expression of Hippo pathway components in the Pontin knockout mice. Using a cellular model we show that Pontin induces YAP activity, YAP nuclear translocation, and transcriptional activity. Our findings identify Pontin as a modulator of adverse cardiac remodelling, possibly via regulation of the Hippo pathway. This study may lead to the development of a new approach to control cardiac remodelling by targeting Pontin.

Cardiac remodelling is a key process in the heart that occurs in response to pathological stimuli. Pathological remodelling is characterized by changes in heart size and shape (hypertrophy and dilatation), the loss of cardiomyocytes due to apoptosis and necrosis, and activation of myofibroblasts leading to fibrosis[1]. If not treated, cardiac remodelling may progress to severe reduction in heart function and eventually heart failure[2].

Reactivation of foetal cardiac genes is often associated with hypertrophic remodelling[3]. These foetal cardiac genes are normally expressed during embryonic heart development, and their expressions diminish in post-natal or adult hearts[4]. Reactivation and re-

expression of these genes in adult hearts during pathological conditions might be the result of activation of signalling pathways that are involved in the cardiac remodelling process[5], such as the Myocyte Enhancer Factor 2 (MEF2)[6], Mitogen Activated Protein Kinases (MAPKs)[7], Wnt/β-catenin[8] and calcineurin-NFAT[9] pathways. Activation of these pathways culminates with the activation of transcription factors which may induce expression of genes, including the foetal cardiac gene programme[5].

Importantly, a number of cardiac foetal genes may play essential roles in mediating cardiac growth during embryogenesis through modulation of myocyte proliferation. Analysis of foetal and adult

[1]Division of Cardiovascular Sciences, Faculty of Biology, Medicine and Health, The University of Manchester, Manchester Academic Health Science Centre, Manchester, United Kingdom. [2]Division of Diabetes, Endocrinology and Gastroenterology, Faculty of Biology, Medicine and Health, The University of Manchester, Manchester Academic Health Science Centre, Manchester, United Kingdom. [3]National Heart and Lung Institute, Imperial College, London, United Kingdom. [4]Present address: Department of Pharmacology, Faculty of Medicine, Universitas Brawijaya, Veteran Street, Malang 65145, Indonesia. [5]These authors contributed equally: Bayu Lestari, Ardiansah Bayu Nugroho, Thuy Anh Bui. ✉e-mail: delvac.oceandy@manchester.ac.uk

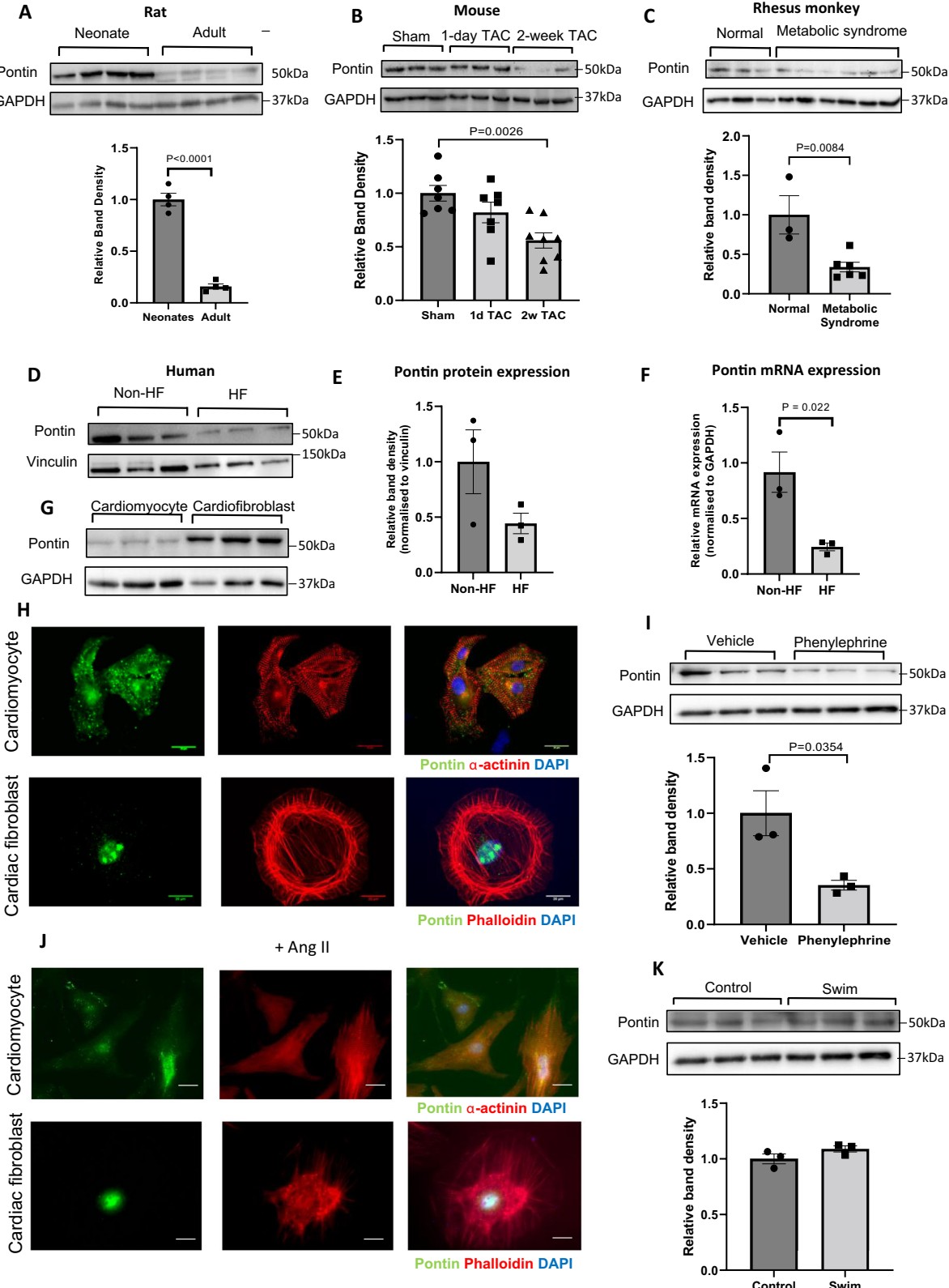

cardiac gene expression profiles has revealed that groups of genes related to cell cycle control and DNA replication are amongst the top differentially regulated genes, with much higher expression in foetal hearts[4]. Modulation of these types of genes in pathological settings may be beneficial in inducing regeneration, as well as controlling remodelling and preventing HF progression[10,11]. Likewise, over-expression of other known cardiac foetal genes such as brain

natriuretic peptide (BNP), may improve cardiac function following stress stimuli[12], whereas hearts lacking atrial natriuretic peptide (ANP) develop more severe cardiomyopathy[13]. Thus, it is essential to study the role of foetal cardiac genes in the adult heart.

A genetic screen analysis in zebrafish has identified a protein complex consisting of Pontin (*Ruvbl1*) and Reptin (*Ruvbl2*)[14]. These proteins are members of the AAA+ (ATPases associated with various

**Fig. 1 | Pontin expression in normal and pathological conditions.**
**A** Representative Western blots and band density analysis of cardiac protein from neonatal compared to adult rat hearts showed significant reduction of expression in adult heart ($n = 4$ in each group). **B** Expression of Pontin is reduced in adult mouse hearts following transverse aortic constriction (sham and 1 d TAC, $n = 7$, 2 d TAC, $n = 8$) and (**C**) in Rhesus monkeys with metabolic syndrome ($n = 6$ in disease group and 3 in normal group). **D** Western blot analysis, (**E**) band density measurement and (**F**) mRNA detection of Pontin expression in human heart samples showed a reduction of its expression in subjects with heart failure ($n = 3$ in each group). **G** Western blot and (**H**) Immunofluorescence detection of Pontin in cultured neonatal rat cardiomyocytes and cardiofibroblasts showing the sub-cellular localization of this protein (scale bars = 20 μm). **I** Expression of Pontin in NRCM was reduced following phenylephrine stimulation ($n = 3$ independent experiments). **J** However, immunofluorescence analysis suggested that there was no change in subcellular localization following stress (scale bars = 20 μm). **K** Expression of Pontin did not change in response to excercise training (swimming) ($n = 3$ mice in each group). Data are presented as mean ± SEM. Statistical tests used: (**A**–**K**) two-tailed Student's $t$-test, (**B**) one-way ANOVA followed by multiple comparisons (Tukey's). For (**H**–**J**) experiments were repeated three times using different batches of primary cells. Source data are provided as a Source Data file.

cellular activities) protein family with various functions including regulation of cell cycle, mitosis, regulation of transcription factors and DNA damage response[15]. In zebrafish these two molecules play an important role in mediating embryonic heart growth[14]. Interestingly, expression of Pontin is elevated in some forms of human cancer[16,17], and it is implicated in regulating cell proliferation in embryonic development[18]. However, despite the knowledge of its crucial role during zebrafish heart development[14], the exact role of Pontin in the adult mammalian heart is unknown. Given the importance of understanding the role of cardiac developmental genes and their potential to lead to the identification of new therapeutic targets to control adverse cardiac remodelling[10,11], in this study we investigated the role of Pontin in cardiomyocytes and in whole hearts. We found a crucial role of Pontin in maintaining cell viability and in modulating pathological hypertrophic growth both basally and in response to stress stimuli, possibly via modulation of the Hippo pathway.

## Results

### Pontin expression in adult heart and in pathological conditions

To study Pontin expression in postnatal mammalian hearts we analysed heart tissues of wildtype (WT) mice and rats. Western blot analysis showed significantly higher expression of Pontin in neonatal (2–3 days old) compared to adult rat hearts (12 weeks old) (Fig. 1A). Pontin expression was also decreased following cardiac stress. In a model of pressure overload hypertrophy that resulted in 1.5 folds increase in heart weight/tibia length ratio as described in our previous publication[19], we found a significant reduction of Pontin expression at 2 weeks after transverse aortic constriction (TAC) in WT mice (Fig. 1B). We also assessed Pontin levels in aged Rhesus monkey hearts, which showed phenotypes of metabolic syndrome including hypertension, high blood glucose and obesity as previously described[20,21]. A significant reduction in Pontin level was observed in the hearts of Rhesus monkeys with metabolic syndrome (Fig. 1C). Consistently, we also found a trend of reduced expression of Pontin in human heart failure (detailed clinical conditions described in Supplementary Table 1) at the protein level (Fig. 1D, E), and at the mRNA level (Fig. 1F).

At a cellular level we detected Pontin expression in both cardiomyocytes and cardiac fibroblasts with higher level of expression in cardiac fibroblasts (Fig. 1G). In cardiomyocytes, Pontin was mainly found in the cytosol with a low level of nuclear localisation, whilst in cardiac fibroblasts Pontin was found in both nucleus and cytosol with highly enriched expression detected within the nucleus (Fig. 1H). Importantly, cardiomyocytes treated with phenylephrine (30 μM, 72 h), which induced cellular hypertrophy by ~1.3 folds as indicated previously[19], displayed significantly reduced Pontin expression compared to vehicle-treated cells (Fig. 1I). However, it seems that Pontin sub-cellular localization was not altered in either cardiomyocytes or cardiofibroblasts following stress (Fig. 1J).

In contrast to the expression in pathological conditions, Pontin expression did not significantly change in exercise-induced cardiac hypertrophy. Western blot analysis of mouse hearts after 4 weeks exercise training (90 min swimming twice a day for 4 weeks), which induced 30% increase in heart weight/body weight ratio[22], did not show any difference in Pontin expression compared to untrained animals (Fig. 1K). Taken together, these data revealed a dynamic control of Pontin expression in the heart upon sustained pathological stimuli, but not in response to physiological stimuli.

### Pontin regulates cardiomyocyte proliferation

The finding that Pontin regulates heart development in zebrafish[14] prompted us to question if Pontin is involved in modulating proliferation of mammalian cardiomyocytes. Therefore, we established a Pontin overexpression system in neonatal rat cardiomyocytes (NRCM) to address this question (Fig. 2A). We then analysed cell proliferation markers, including EdU incorporation rate, Ki67 and phospho-Histone H3 (pHH3) expressions. Our observations showed that Pontin overexpression increased NRCM proliferation (Fig. 2B–E and Supplementary Fig. 1A–C). Consistently, when we inhibited Pontin expression using siRNA (Fig. 2F) we found a significant reduction in cardiomyocyte proliferation (Fig. 2G–J and Supplementary Fig. 2A–C).

### Pontin regulates Angiotensin-induced hypertrophic response

We followed up the finding above by addressing the question whether Pontin regulates hypertrophic growth in addition to hyperplasia. Basally, there was no difference in cardiomyocyte size following Pontin overexpression or gene silencing. However, in response to Angiotensin II (Ang II) treatment (1 μM, 48 h) Pontin overexpression resulted in a reduced Ang II-mediated hypertrophic response, while Pontin knock-down exaggerated the hypertrophic response as indicated by cell size measurement (Supplementary Fig. 3A–D). Consistently, analysis using BNP-luciferase reporter revealed a lower BNP-luciferase signal in Pontin overexpressing cells but higher BNP-luciferase signal in NRCM lacking Pontin following Ang II stimulation (Supplementary Fig. 3E, F), suggesting a possible role for Pontin in mediating hypertrophic growth.

### Pontin protects against apoptosis and oxidative stress

In addition to inducing hypertrophic growth, Ang-II induces apoptosis through activation of both AT1 and AT2 receptors[23]. To evaluate if Ang II-dependent apoptosis is affected by Pontin, we conducted TUNEL assay in NRCM overexpressing or lacking Pontin following Ang II stimulation. We found that the apoptosis level was significantly reduced in cells overexpressing Pontin (Fig. 3A, B and Supplementary Fig. 4A). In contrast, we detected a higher number of apoptotic cells in NRCM deficient of Pontin (Fig. 3C, D and Supplementary Fig. 4B).

In cardiomyocytes, AngII-induced cell death may be triggered by increased oxidative stress due to NADPH oxidase activation and reactive oxygen species (ROS) production in the mitochondria[24]. We therefore performed dihydroethidium (DHE) staining on NRCM to assess the level of ROS production in Pontin knock-down and overexpressing cardiomyocytes. In keeping with previous data we observed a protective role of Pontin overexpression in reducing ROS levels following Ang-II treatment, whereas knock down of this gene resulted in a higher level of intracellular ROS (Fig. 3E–H). We followed up this finding by examining the level of cell viability in response to oxidative stress induced by peroxide treatment (200 μM, 4 h). Consistently, Pontin overexpression increased cell viability following oxidative stress induced by peroxide treatment, whereas Pontin

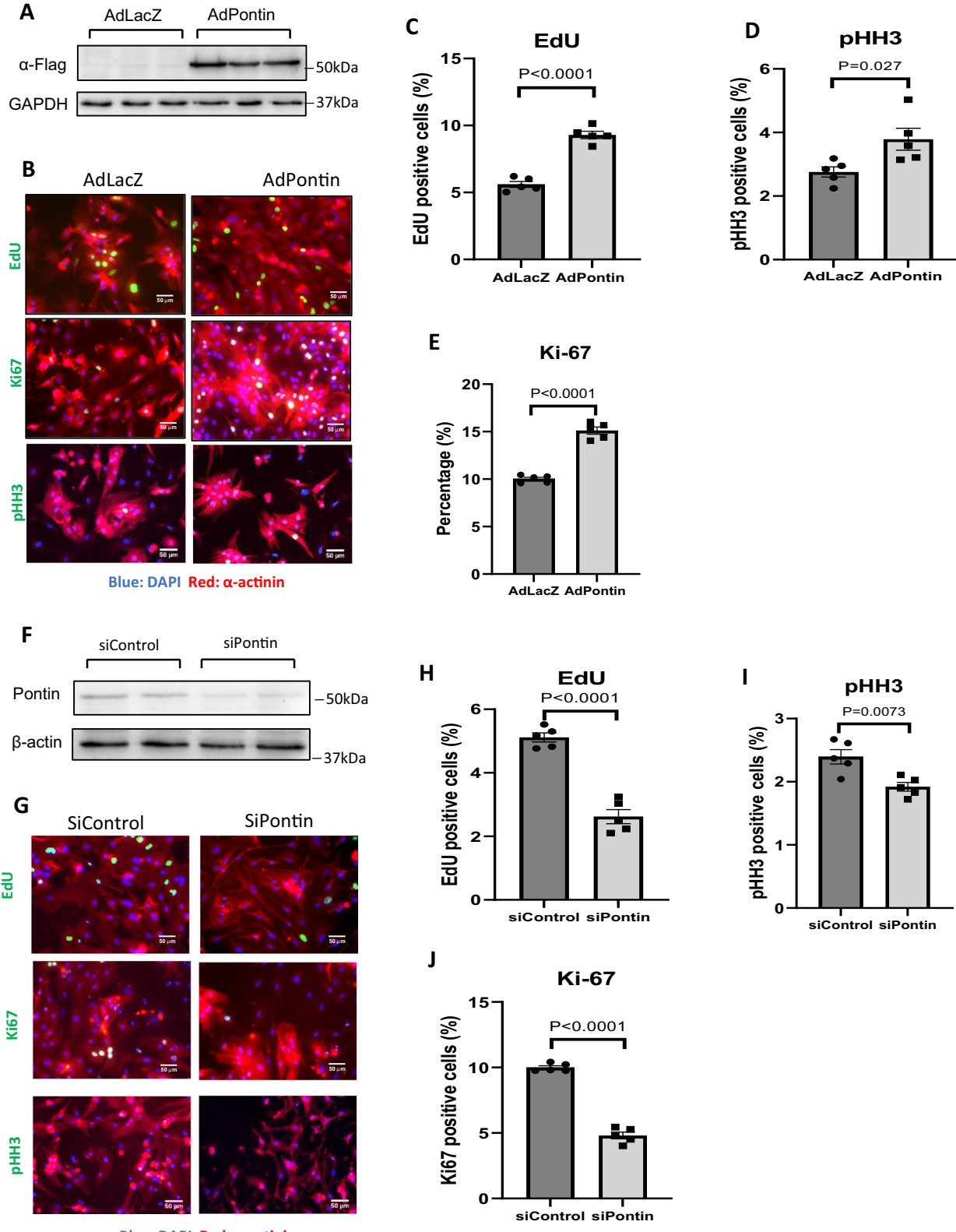

**Fig. 2 | Pontin regulates proliferation of neonatal rat cardiomyocytes. A** Representative Western blot image showed enhanced Pontin expression following transduction with adenovirus carrying *Pontin* cDNA ($n = 3$ independent experiments in each group). **B** Images of EdU incorporation assay, Ki67and pHH3 detection in NRCM overexpressing Pontin compared to control (scale bars = 50 μm). Images of separated channels can be seen in Supplementary Fig. 1A–C. **C** Quantification of EdU, (**D**) pHH3 and (**E**) Ki67 positive cells suggested that Pontin overexpression significantly enhanced NRCM proliferation ($n = 5$ independent cell preps with minimum of 3 replications in each prep). **F** Example of Western blot showing reduction of Pontin after siRNA transfection. **G** Fluorescent images of EdU incorporation assay, Ki67and pHH3 detection in NRCM deficient of Pontin compared to control (scale bars = 50 μm). Images of separated channels can be seen in Fig. S2A–C. Measurement of (**H**) EdU, (**I**) pHH3 and (**J**) Ki67 positive nuclei indicated that Pontin deficiency led to a reduction in NRCM proliferation ($n = 5$ independent cell preps with minimum of 3 replications in each prep). Data are presented as mean ± SEM. Statistical tests used are two-tailed Student's *t*-test for (**C**–**J**). Source data are provided as a Source Data file.

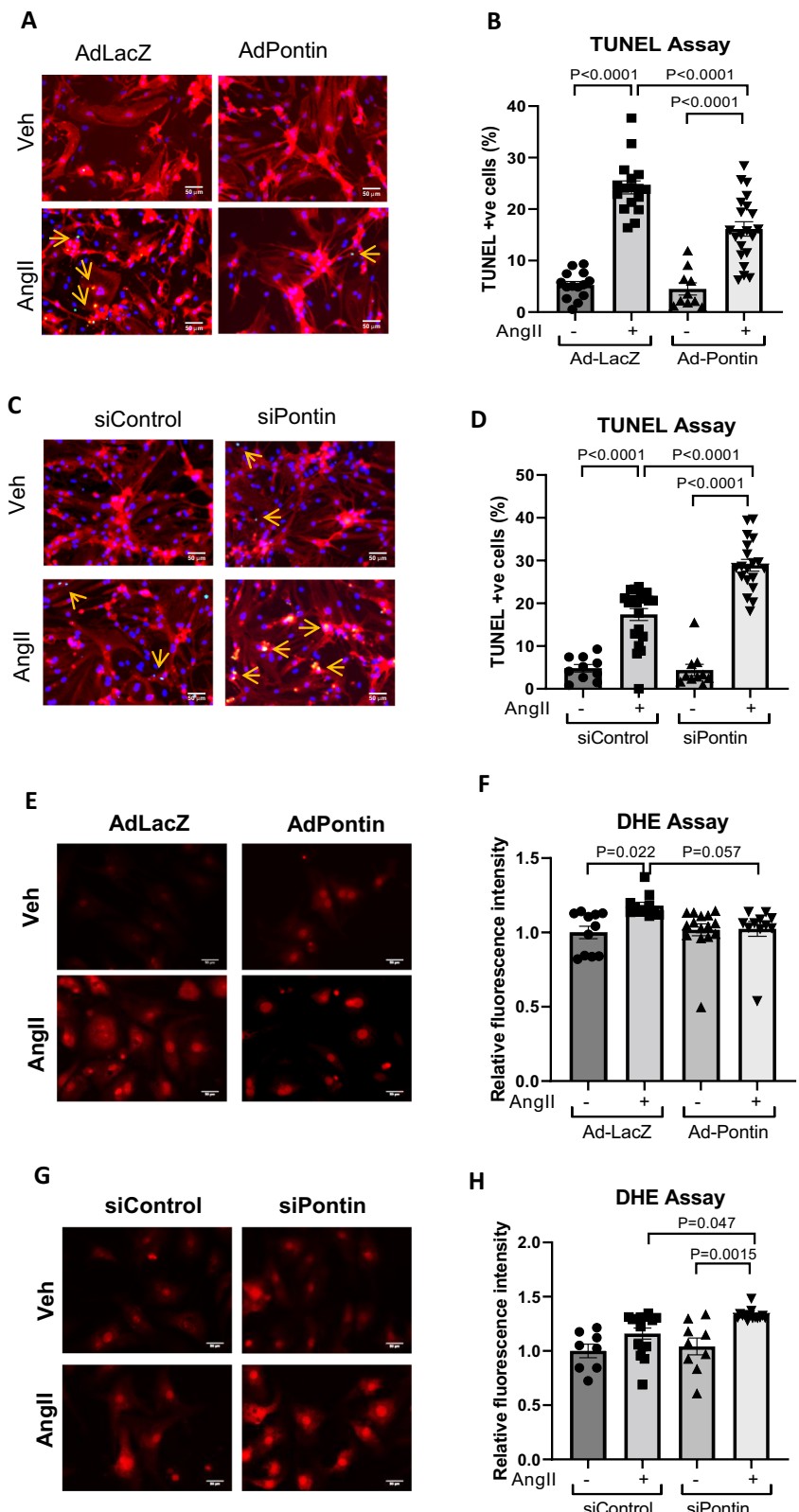

knockdown led to a reduction in NRCM viability (Supplementary Fig. 5A, B).

**Pontin knock out results in severe cardiomyopathy in vivo**
Our data above indicate that Pontin may play essential role in mediating cardiomyocyte growth and survival, therefore it is important to understand Pontin's role in the heart in vivo. To address this question we generated an inducible cardiomyocyte specific knockout of Pontin in mice (Pontin$^{icKO}$) using the αMHC-MerCreMer (αMCM)/loxP system. Details of the generation of Pontin$^{icKO}$ mice are described in supplementary fig. 6 and in the methods section. We used αMCM-Pontin$^{floxed}$ mice without tamoxifen injection and αMCM TG mice (with and without tamoxifen treatment) as controls.

**Fig. 3 | Pontin mediates apoptosis and oxidative stress following angiotensin II stimulation. A** Representative images of TUNEL assay and (**B**) quantification of TUNEL positive cells in NRCM overexpressing Pontin vs control following treatment with Ang II (1 μM, 48 h) (AdLacZ, $n = 14$ replications, AdlacZ+Ang II, $n = 17$ replications, AdPontin, $n = 10$ replications, AdPontin+Ang II, $n = 21$ replications).Images of separated channels from TUNEL experiments can be seen in Supplementary Fig. 4A, B. **C** TUNEL assay and (**D**) quantification of TUNEL positive cells in NRCM with Pontin gene silencing following Ang II stimulation (siControl, $n = 10$ replications, siControl+Ang II, $n = 21$ replications, siPontin, $n = 10$ replications,

siPontin+Ang II, $n = 19$ replications). **E** Images of DHE assay and (**F**) measurement of fluorescent intensity in NRCM overexpressing Pontin ($n = 11$ replications each for AdLacZ, AdLacZ+Ang II and AdPontin+Ang II, for AdPontin, $n = 15$ replications) and (**G, H**) NRCM lacking Pontin (siControl, $n = 8$ replications, siControl+Ang II, $n = 15$, siPontin, $n = 9$, siPontin+Ang II, $n = 15$) following Ang II stimulation. Overall, the data indicated that Pontin expression was protective against apoptosis and oxidative stress due to Ang II stimulation. Scale bars = 50 μm. Data are presented as mean ± SEM. Statistical tests used: (**B–H**) one-way ANOVA followed by multiple comparisons test (Tukey's). Source data are provided as a Source Data file.

Pontin inducible cardiomyocyte knockout (Pontin[icKO]) was achieved by treating αMCM-Pontin[floxed] mice with a single injection of 40 μg/kg BW tamoxifen intraperitoneally. Western blot data confirmed reduction of Pontin expression by ~50-60% in cardiomyocytes isolated from adult Pontin[icKO] mice (Fig. 4A), whereas Pontin expression in cardiac fibroblasts was unaltered (Fig. 4B) indicating cell specific ablation. We then performed serial echocardiography analysis at 1, 3 and 4 weeks post-tamoxifen injection. We found a marked reduction of cardiac function (ejection fraction) in Pontin[icKO] mice at 4 weeks post tamoxifen injection (Fig. 4C). Interestingly, ejection fraction was not reduced at 1 and 3 weeks post-tamoxifen indicating an abrupt deterioration between week 3 and 4. Analysis of echocardiography parameters suggested LV chamber dilatation and significant reduction of wall thickness in systole (Supplementary Fig. 7).

**Pontin knockout increases hypertrophy, apoptosis and fibrosis**
To understand the pathological changes in Pontin[icKO] hearts, we sacrificed mice at day 24 post-tamoxifen injection and analysed heart tissues at this time point. We found enlarged hearts in Pontin[icKO] mice compared to controls as indicated by heart weight/body weight (HW/BW) ratio (Fig. 4D). Cardiomyocyte cross-sectional area of Pontin[icKO] mice were also larger than controls (Fig. 4E, F). Moreover, expression of hypertrophic marker (ANP) was also higher in Pontin[icKO] hearts (Fig. 4G).

We then performed experiments to analyse the level of fibrosis and apoptosis. Assessment of fibrotic area from Masson's trichrome stained tissue sections and analysis of Col1a expression strongly indicated an increased level of fibrosis in Pontin[icKO] mice (Fig. 4H–J). Importantly, TUNEL assays suggested that ablation of Pontin in adult hearts resulted in increased cardiomyocyte apoptosis (Fig. 4K, L). Moreover, analysis of the expression of major regulators of apoptosis showed that expressions of Bad, Bax and Caspase-3, which are apoptosis inducers, were markedly increased in Pontin[icKO] mice, whilst expression of p53 and Bcl-xL, were not altered (Supplementary Fig. 8A–F).

Together, the data suggested that ablation of Pontin in adult cardiomyocytes induced severe adverse remodelling, which was characterized by cardiomyocyte hypertrophy, apoptosis and fibrosis.

**Pontin knockout exaggerated the effects of Ang-II stimulation**
Our in vitro data suggested that genetic silencing of Pontin exaggerates the effects of αMG-II stimulation. We therefore investigated the effects of Ang-II stimulation in Pontin[icKO] mice. Ang-II was infused at a dose of 1.5 mg/kg BW/ day using mini-osmotic pump started at day 7 after tamoxifen injection. Echocardiography analysis showed that Pontin[icKO] mice exhibited a severe reduction of cardiac function at day 4 post Ang-II infusion. Therefore, experiments were terminated at day 3 and the endpoint phenotype analysis was performed at day 3 after mini-pump implantation. We observed a significant reduction of EF in Pontin[icKO] mice compared to controls (Supplementary Fig. 9A), although the hypertrophic response (HW/BW ratio, cardiomyocyte size and ANP expression) and fibrosis (Masson's trichrome staining and collagen 1 expression) were not different at this time point (Supplementary Fig. 9B–H). Consistent with the basal phenotype, apoptosis level was increased in Pontin[icKO] mice at 3 days after Ang-II induction

(Supplementary Fig. 9I, J). Overall, the data indicated that Pontin deletion in cardiomyocytes accelerates reduction of cardiac contractility and the increase in apoptosis in response to pathological stimuli.

**RNASeq analysis of transcriptomic pattern of Pontin[icKO] hearts**
RNAseq analysis on Pontin[icKO] hearts was performed to gain insight into the mechanism. Total RNA was examined at 1 and 3 weeks after tamoxifen injection to analyse changes in gene expression pattern at the early and late stages after Pontin deletion. At 1 week time point we found 60 upregulated and 47 downregulated genes with FDR values <0.05 and $\log_2$ fold change >1 or <-1, whereas at 3 weeks we found 916 genes were upregulated and 1051 genes were downregulated (Fig. 5A, B). Of all differentially expressed genes (DEGs), 60 were differentially regulated at both 1 and 3 weeks after tamoxifen (Supplementary Fig. 10A). To describe the functions of these genes we used gene ontology analysis. Sets of genes that were most significantly different at both time points were genes related to the regulation of cell cycle and cell division (Supplementary Fig. 10B).

Furthermore, ingenuity pathway analysis (IPA) of the DEGs showed enrichment of genes associated with cellular development, cell growth and proliferation, cell cycle and cell death/survival (Fig. 5C, D). This finding prompted us to analyse the activity of several major signalling pathways commonly involved in regulating cardiomyocyte growth, hypertrophy, proliferation and apoptosis.

We used the H9c2 cardiomyoblast cell line for this purpose. H9c2 cells were transfected with luciferase constructs reporting the activity of either β-catenin (Wnt pathway), NFκB, NFAT, YAP, STAT3 or AP1 signalling pathways, all of which have been known to regulate cardiomyocyte growth, proliferation and/or survival. After 24 h, cells were treated with adenovirus expressing Pontin or siRNA to knockdown Pontin. The luciferase activity was measured 48 h later. As shown in Fig. 5E, F we found that the activity of YAP, which is the main effector of the Hippo pathway, was significantly induced in cells overexpressing Pontin but reduced in cells lacking Pontin. NFAT-luciferase activity was also differentially regulated but only in the Pontin knockdown model.

We next performed gene set enrichment analysis (GSEA) to further assess the involvement of the Hippo pathway in Pontin knockout. We used a canonical pathway database and found a significant association between Pontin deletion and expression of Hippo pathway components (Fig. 5G, H).

**Expression of Hippo pathway components were altered in Pontin[icKO] mice**
Based on the KEGG database[25] we selected genes that are known as the core components, the downstream effectors, positive and negative modulators of the pathway (Fig. 6A). As indicated by the heat map we observed a trend of increased expression of the core components of the Hippo pathway and the negative regulators of YAP/TAZ in Pontin[icKO] mice, in particular at 3 weeks after tamoxifen injection (Fig. 6B). Individual analysis of selected genes such as MST1, SAV1 and MOB1, which are members of the core components of the Hippo pathway suggested a significant increase in mRNA levels at 3 weeks after tamoxifen (Fig. 6C–E).

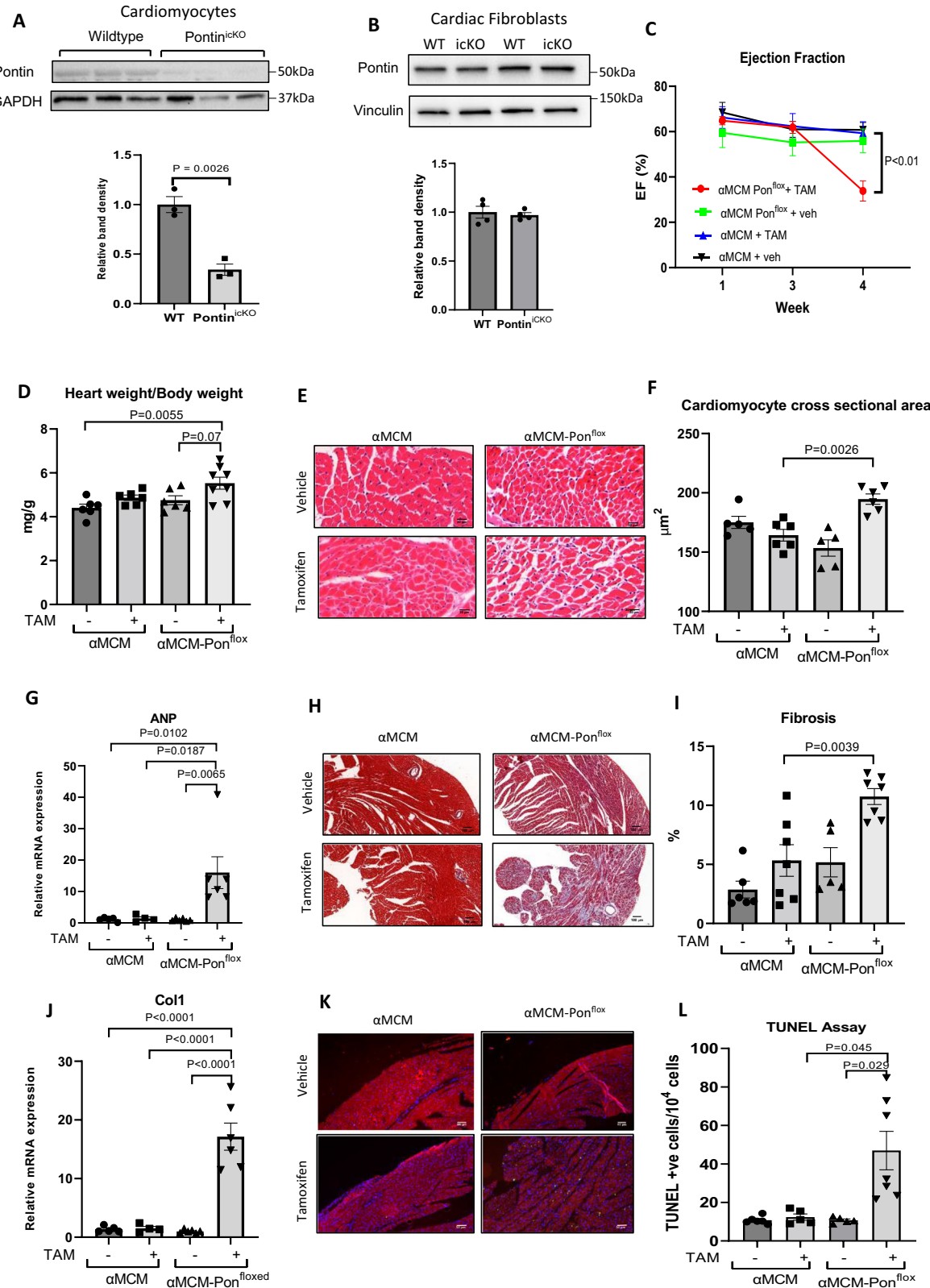

To follow up the RNASeq data we went on to examine protein expression of several members of the Hippo pathway in mouse heart. Western blot analysis confirmed increased expressions of MST1 and MOB1 in Pontin[icKO] hearts at protein level, whilst expressions of LATS1 and YAP were not altered (Fig. 6F–J). To examine if the increase expressions of Hippo pathway components affected YAP activation, we conducted immunohistochemistry analysis on cardiac tissue

sections of Pontin[icKO] mice. We observed a trend of increased YAP phosphorylation in Pontin[icKO] heart sections, although it did not reach statistical significance (Fig. 6K, L). This data suggested a possible reduction of YAP activity in Pontin[icKO] hearts. Since Hippo pathway and YAP are strongly associated with the regulation of apoptosis, this finding may explain the cardiac phenotypes that we observed in Pontin[icKO] mice.

**Fig. 4 | Pontin inducible cardiomyocyte specific knockout in mice results in severe cardiomyopathy. A** Representative Western blot image of Pontin expression in cardiomyocytes ($n = 3$ mice in each group) and (**B**) cardiac fibroblasts from Pontin[icKO] mice and WT controls, and quantification of band density ($n = 4$ mice in each group). **C** Serial echocardiography analysis was performed to asses cardiac function of Pontin[icKO] mice. Left ventricular ejection fraction (EF) was evaluated at 1,3 and 4 weeks after tamoxifen injection. Results showed significant reduction of EF in Pontin[icKO] mice at 4 weeks after tamoxifen injection ($n = 6$ mice for αMCM, αMCM+tamoxifen (TAM) and αMCM Pon[flox], $n = 8$ mice for Pontin[icKO] αMCM Pon[flox] + TAM). **D** Analysis of Heart weight/body weight ratio indicated bigger heart size in Pontin[icKO] mice ($n = 6$ mice for αMCM, αMCM+tamoxifen (TAM) and αMCM Pon[flox], $n = 8$ mice for Pontin[icKO] αMCM Pon[flox] + TAM). **E** Representative images of cardiac tissue section stained with hematoxyllin and eosin (scale bars = 20 μm) and (**F**) quantification of cardiomyocyte cross sectional area showing larger cardiomyocyte size in Pontin[icKO] mice ($n = 5$ mice for αMCM and αMCM Pon[flox], $n = 6$ mice

for αMCM + TAM and αMCM Pon[flox] + TAM). **G** qPCR analysis of ANP mRNA level suggested higher level of cardiac ANP in Pontin[icKO] ($n = 5$ mice for αMCM and αMCM Pon[flox] + TAM, $n = 4$ mice for αMCM + TAM, $n = 6$ mice for αMCM Pon[flox]). **H** Examples of Masson's trichrome stained cardiac tissue sections (scale bars = 100 μm), (**I**) quantification of fibrotic area ($n = 6$ mice for αMCM, $n = 7$ mice for αMCM + TAM and αMCM Pon[flox] + TAM, $n = 5$ mice for αMCM Pon[flox]) and (**J**) qPCR analysis of collagen I level ($n = 5$ mice for αMCM and αMCM Pon[flox] + TAM, $n = 4$ mice for αMCM + TAM, $n = 6$ mice for αMCM Pon[flox]) indicated a significant increase of cardiac fibrosis in Pontin[icKO] mice. **K** TUNEL staning followed by (**L**) quantification of TUNEL positive cells demonstrated enhanced apoptosis following Pontin deletion in cardiomyocytes in vivo (scale bars = 50 μm)($n = 6$ mice for αMCM, $n = 5$ mice for αMCM + TAM and αMCM Pon[flox], $n = 7$ mice for αMCM Pon[flox] + TAM). Data are presented as mean ± SEM. Statistical tests used: (**A**, **B**) two-tailed Student's *t*-test, (**C**–**L**) one-way ANOVA followed by multiple comparisons test (Tukey's). Source data are provided as a Source Data file.

## Pontin regulates Hippo pathway in cardiomyocytes

To further analyse the mechanism as to how Pontin regulates the Hippo pathway we performed experiments in NRCM. We used a luciferase reporter to monitor YAP activity in NRCM and detected significantly higher YAP-luciferase signal in NRCM overexpressing Pontin (Fig. 7A). In contrast, YAP-luciferase activity was reduced in NRCM lacking Pontin (Fig. 7B). Consistently, reintroduction of Pontin expression in NRCM-lacking Pontin rescued YAP activity (Supplementary Fig. 11A). Furthermore, analysis of YAP nuclear translocation using a GFP-YAP construct showed an elevated number of YAP positive nuclei in Pontin overexpressing cells, whereas NRCM deficient of Pontin displayed lower levels of nuclear YAP (Fig. 7C–F). Importantly, Western blot analysis showed increased active YAP and reduction of phospho-YAP following Pontin overexpression (Fig. 7G–I) whereas Pontin knockdown reduced the active YAP level and increased phospho-YAP (Fig. 7J–L).

To ascertain that YAP regulation is the mechanism responsible for the phenotypes of Pontin downregulation, we conducted YAP rescue experiments by using inducible-expression system of constitutively active YAP[S127A] that we have developed previously[26]. YAP[S127A] expression in Pontin deficient cardiomyocytes was induced by doxycycline treatment. As shown in Fig. 7M the reduction of YAP-luciferase signal due to Pontin knockdown was rescued by expression of active YAP. Moreover, expression of YAP[S127A] also rescued the reduction of cardiomyocyte proliferation due to Pontin knockdown as indicated by Ki67 detection (Supplementary Fig. 11B, C). Together, these data strongly indicate that the cardiomyocytes phenotypes observed following Pontin knock down are mainly mediated by YAP.

In keeping with these data, expression of YAP target genes such as *Birc5*, *Cyr61*, *Fgf2*, *Pik3cb* and *Tead4* were significantly reduced in NRCM lacking Pontin (Supplementary Fig. 11D). However, we did not observe any significant changes in the expression of YAP target genes in Pontin overexpressing cells (Supplementary Fig. 11E).

Since the main finding in the RNAseq analysis was the increased expression of Hippo core components we then examined expression as well as activation (phosphorylation) of core components of Hippo pathway in NRCM overexpressing or lacking Pontin. Interestingly, contrary to the finding in knockout mice, we did not observe any changes in the expression level of Hippo pathway components. However, we found reduced phosphorylation level of MOB1 in Pontin overexpression cells and higher MOB1 phosphorylation in cells lacking Pontin (Fig. 7N–Q), which in turn may contribute to YAP regulation.

To analyse if Pontin also modulates YAP activation in adult cardiomyocytes, we isolated and cultured adult rat cardiomyocytes (ARCM). Pontin was then overexpressed in ARCM and YAP activity was measured. Pontin overexpression significantly induced YAP-luciferase activity and YAP nuclear translocation in ARCM (Fig. 8A–C). Moreover, ARCM overexpressing Pontin displayed increase proliferation rate as indicated by expression of Ki67 (Fig. 8D, E). Together, these data

suggests that the phenotypes due to Pontin modulation in neonatal cells are recapitulated in adult cardiomyocytes.

To further understand the mechanism of MOB1 regulation by Pontin, we generated an ATPase-deficient mutant of Pontin (Pontin[D302N]). We overexpressed this protein in NRCM and found that Pontin[D302N] failed to reduce MOB1 phosphorylation (Fig. 8F, G), suggesting that the regulation of MOB1 depends on Pontin activity and confirmed the direct link between Pontin and MOB1 regulation.

Since we found that in Pontin[icKO] mice the expressions of MST1 and MOB1 were increased (Fig. 6C–F) we sought to understand whether inhibition of these proteins could rescue the phenotype of Pontin-deficient cardiomyocytes. Using NRCM lacking Pontin (by siRNA transfection) we observed that MST1 pharmacological inhibition, MOB1 gene silencing and YAP activation rescued the reduction of YAP activity due to Pontin deletion (Fig. 8H–J). These results further supported the idea that Pontin exerts its function by regulating components of the Hippo pathway.

## Pontin overexpression ameliorates the pathological effects of Ang-II

Since deletion of Pontin in cardiomyocytes resulted in severe adverse effects both basally and following stress, the logical hypothesis would be that overexpression of Pontin in adult cardiomyocytes will produce beneficial and protective effects against pathological stimuli. As an initial step to test this hypothesis we generated transgenic mice with cardiomyocyte specific overexpression of Pontin (Pontin[cTG]). Pontin expression was driven by the αMHC promoter. Western blot analysis as shown in Fig. 9A and Supplementary Fig. 12A confirmed the cardiac specific overexpression of Pontin in these mice.

There was no significant difference in terms of basal cardiac phenotype between Pontin[cTG] and control littermates. However, in response to Ang-II infusion (1.5 mg/kg BW/day for 2 weeks) we found that Pontin[cTG] mice displayed reduced hypertrophy compared to non-TG littermates, as indicated by HW/BW analysis, despite a comparable rise in blood pressure between the groups (Fig. 9B, C). Moreover, cardiomyocyte size, ANP and BNP expression were also reduced in Pontin[cTG] mice, confirming the protective effect of Pontin expression against AngII-induced hypertrophy (Fig. 9D–G). Fibrosis level was also reduced in Pontin[cTG] mice (Fig. 9H–J). Equally important, analysis of apoptosis (Fig. 9K, L) and ROS level (Supplementary Fig. 12B, C) showed consistent protective effects of Pontin expression. Further analysis of echocardiographic parameters confirmed the reduction in hypertrophic response as indicated by smaller wall thickness at diastole in Pontin[cTG] mice (Supplementary Fig. 12D–I). Overall, consistent with our findings in knockout model, we found that overexpression of Pontin in adult cardiomyocytes might be beneficial in protecting the heart against adverse remodelling due to chronic Ang-II stimulation.

We observed a trend of reduced expression of endogenous Pontin in WT mice after Ang-II treatment (Supplementary Fig. 13A), which is

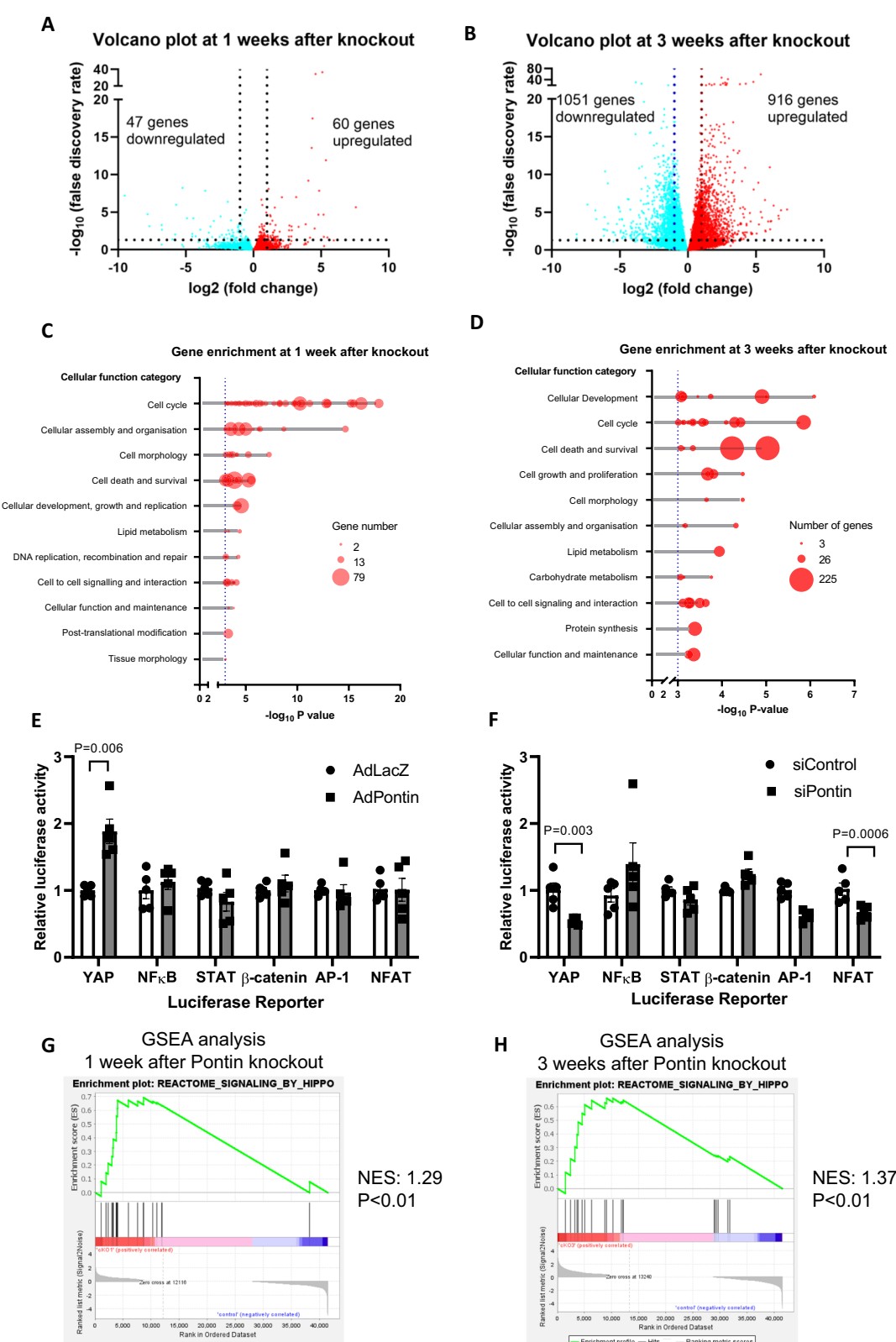

**Fig. 5 | Transcriptomic and pathway screening analysis identifies Hippo pathway to be differentially regulated following Pontin knockout.** **A** Volcano plot of differentially expressed genes (DEGs) in Pontin[ickO] hearts compared to controls at 1 week and (**B**) 3 weeks following tamoxifen injection (*n* = 3 mice in each group). **C** Ingenuity pathway analysis (IPA) of cellular functions of DEGs at 1 week after Pontin knockout and (**D**) at 3 weeks after Pontin knockout. The enriched cellular functions are written in *Y*-axis. The *X*-axis represents negative log of *P*-values. The size of the dots represents the number of DEGs in each category. **E** Results of

pathway screening assays in H9c2 cardiac myoblast cell lines using Pontin overexpression and (**F**) Pontin silencing systems demonstrated a consistent modulation of YAP activity by Pontin (*n* = 5 independent experiments in each group). **G** Gene set enrichment analysis (GSEA) - enrichment plot of gene set 'Hippo signalling pathway' at 1 week after Pontin knock out and (**H**) at 3 weeks after knockout indicating significant enrichment of this gene set following Pontin knockout. Data are presented as mean ± SEM. Statistical significance was determined by multiple two-tailed Student's *t*-test (E&F). Source data are provided as a Source Data file.

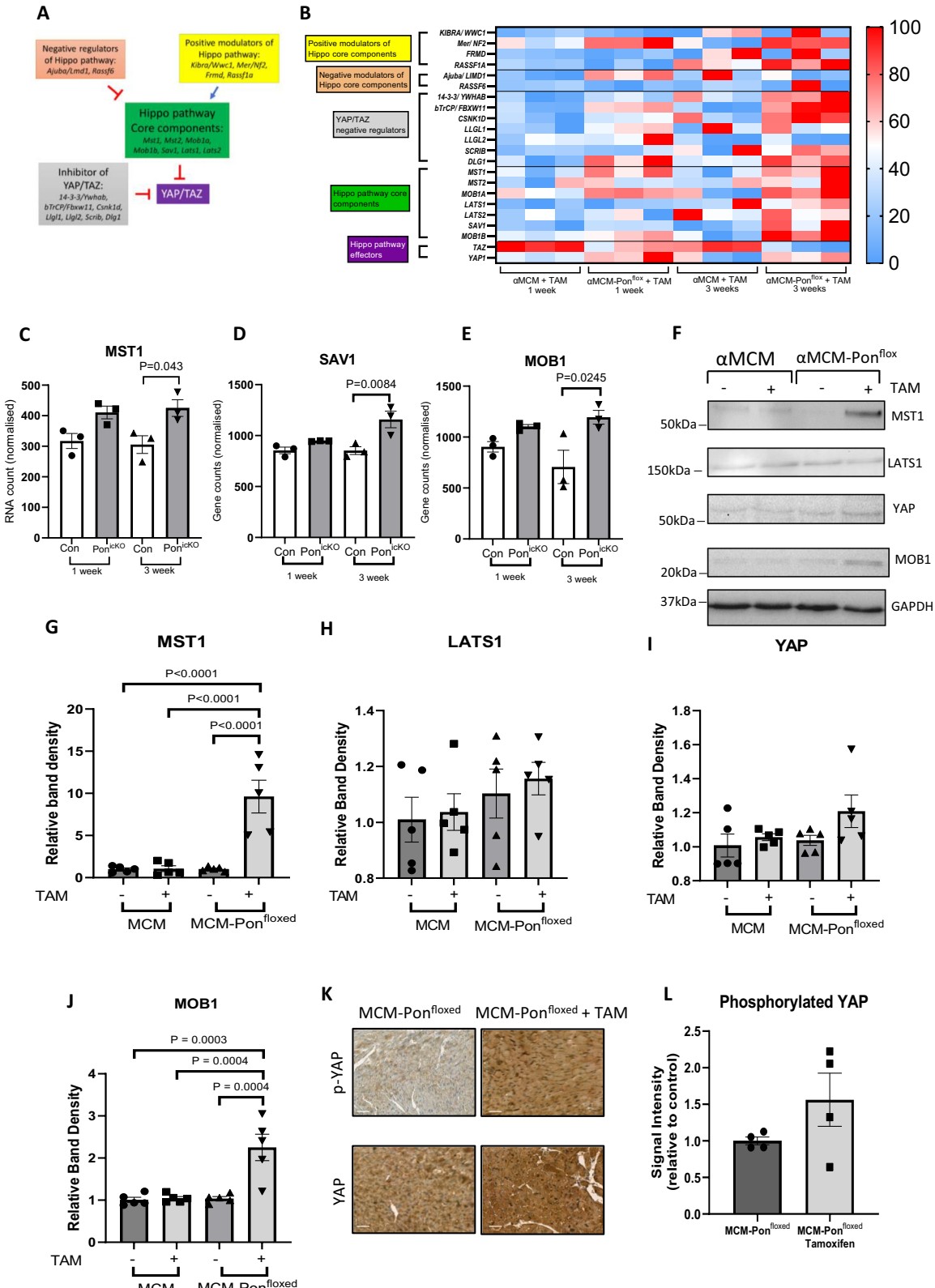

consistent with the findings that cardiac Pontin was downregulated in response to various pathological stress (see Fig. 1). However, expression of total Pontin in Pontin[cTG] mice (endogenous plus transgene) was not affected by Ang-II treatment (Supplementary Fig. 13B), which probably explains why the Pontin[cTG] mice are protected against adverse remodelling. Consistently, Pontin[cTG] mice displayed significantly reduced levels of YAP phosphorylation compared to WT,

indicating higher YAP activity at basal condition. Interestingly, while phospho-YAP level was reduced in WT mice after Ang-II stimulation, it did not change in Pontin[cTG] mice following Ang-II treatment (Supplementary Fig. 13C). In keeping with these findings, we also found reduced level of phospho-MOB1 in Pontin[cTG] mice (Supplementary Fig. 13D), further supporting the notion of Hippo pathway regulation by Pontin in Pontin[cTG] mice.

**Fig. 6 | Expression analysis of cardiac transcripts and proteins from WT and Pontin<sup>icKO</sup> mice revealed differential expression of members of Hippo pathway. A** Selection and classification of Hippo pathway components based on signalling pathway database (KEGG database). Genes are classified as Hippo pathway core components, negative regulator of Hippo pathway, positive regulator of Hippo pathway, Hippo pathway effector (YAP/TAZ) and inhibitor of YAP/TAZ. **B** Heat map showing cardiac expression of Hippo pathway components in controls vs Pointin<sup>icKO</sup> mice at 1 week and 3 weeks after Pontin deletion (*n* = 3 mice in each group). The data indicated an increased in the expression of core Hippo components in Pontin<sup>icKO</sup> hearts partularly at 3 weeks after tamoxifen injection. Furthermore, individual transcript analysis of key components of Hippo pathway revealed that the expression of (**C**) MST1, (**D**) SAV1 and (**E**) MOB1 were significantly upregulated at 3 weeks after Pontin deletion in cardiomyocytes (*n* = 3 mice in each

group). **F** Representative Western blots to assess protein expression of key components of Hippo pathway in Pontin<sup>icKO</sup> (αMCM-Pon<sup>flox</sup> + tamoxifen) vs controls (the other 3 groups). Analysis of band density of (**G**) MST1, (**H**) LATS1, (**I**) YAP and (**J**) MOB1 indicated a significant increase in the expression of MST1 and MOB1 in Pontin<sup>icKO</sup> mice (*n* = 5 mice in each group). **K** Representative images of immunohistochemistry analysis to detect phospho-YAP and total YAP in heart tissue sections of Pontin<sup>icKO</sup> (MCM-Pon<sup>floxed</sup> + TAM) and control mice (MCM-Pon<sup>floxed</sup> without tamoxifen)(scale bars = 20μm). **L** Quantification of signal intensity indicated a trend of increased phospho-YAP activity in Pontin<sup>icKO</sup> mice (*n* = 4 mice in each group). Data are presented as mean ± SEM. Statistical tests used: (**C–L**) one-way ANOVA followed by multiple comparisons test (Tukey's), (**L**) Student's *t*-test. Source data are provided as a Source Data file.

## Analysis of transcriptomic profile of Pontin<sup>cTG</sup> mice

To understand the mechanism, RNAseq analyses were conducted comparing the transcriptomic profile between Pontin<sup>cTG</sup> and WT littermates after 2 weeks of Ang-II infusion. IPA analysis of the DEGs indicated enrichments of genes related to cellular movement, cell morphology, cell growth and proliferation, cell development, cell cycle, cell death and survival, organ development and morphology (Fig. 10A, B). Importantly, GSEA analysis identified Hippo signalling pathway as the most significant pathway that was differently regulated based on NES and FDR values (Fig. 10C, D). Consistently, IPA analysis of transcription regulators showed that TEAD, which is the main effector of Hippo/YAP pathway, was the most significant transcription factor that was activated in Pontin<sup>cTG</sup> mice (Fig. 10E). Analysis of individual genes within the Hippo pathway core components revealed that expressions of MOB1 and LATS1 were significantly downregulated in Pontin<sup>cTG</sup> mice, whereas there was a trend of downregulation of MST1, but not statistically significant (Fig. 10F–H). Overall, the RNAseq data of Pontin<sup>cTG</sup> was consistent with those of Pontin<sup>icKO</sup> confirming that regulation of Hippo pathway might be the mechanism underlying the cardiac phenotype of these mice.

## Discussion

The main finding of this study is that a member of the AAA+ ATPase family, Pontin (*Ruvbl1*), regulates cardiomyocyte growth and survival in adult mouse hearts. Pontin expression was reduced in various pathological conditions, such as pressure overload, adrenergic stimulation, metabolic stress and importantly in human heart failure. Interestingly, our findings demonstrate that re-expression of Pontin in adult cardiomyocytes is beneficial in protecting the heart against apoptosis, fibrosis and oxidative stress following Ang-II stimulation.

A set of foetal genes, which are expressed highly during heart development, play a crucial role in promoting cellular growth and differentiation while maintaining cardiomyocyte survival[5,27]. Expression of these genes is suppressed and becomes silent as the postnatal heart adapts to the new environment[28]. However, a number of studies showed that these foetal genes are reactivated during stress (reviewed in ref. [29]). The reason why these genes are reactivated in pathological conditions and the mechanism underlying it are not completely understood, however, some foetal genes might have detrimental effects if overexpressed in cardiomyocytes. For example, overexpression of the foetal gene β-myosin heavy chain in adult mouse heart resulted in increased hypertrophy and reduced function after adrenergic stimulation[30]. Conversely, modulation of some particular types of developmental genes may produce beneficial effects. Activation of YAP, the main downstream effector of the Hippo pathway, resulted in the induction of cardiomyocyte proliferation, sufficient to reduce the scar size, as well as improve cardiac function following MI[31]. Consistently, inhibition of upstream components of the pathway such as MST1/2 and Sav1, which results in YAP activation, improved cardiac phenotype in models of MI, pressure overload and heart failure[10,32].

Thus, cardiac developmental genes may become an attractive therapeutic target if modulated correctly.

Pontin is known as a gene that is highly expressed during heart development and tightly involved in regulating foetal cardiac growth[14]. In the present study, we demonstrate key roles of Pontin in mammalian cardiomyocytes and adult hearts. By using gene ablation and overexpression approaches, we confirmed that Pontin regulates key processes in cardiomyocytes, including proliferation, apoptosis/survival, and hypertrophic response induced by angiotensin. Pontin inducible cardiac-specific knock-out mice displayed a severe cardiomyopathy phenotype characterized by profound apoptosis, hypertrophy, fibrosis and reduction of contractile function. This phenotype was accentuated following angiotensin infusion. In contrast, cardiac-specific Pontin overexpression protected mice from pathological remodelling following angiotensin treatment. Taken together, despite its lower expression in the postnatal heart, these findings suggest that Pontin is essential to preserve cellular processes in cardiomyocytes and protects them against pathological insult.

Our findings provide additional evidence on the cellular role of Pontin. In other cell types, Pontin regulates cell cycle/mitotic progression[33,34], chromatin remodelling and transcription regulation[35], nuclear regulation of autophagy through epigenetic mechanism[36], DNA damage response[37], and maintaining ES cell pluripotency[38], indicating its involvement in various biological processes. In terms of Pontin's role in pathological conditions, perhaps the most significant findings were its role in cancer development. Pontin was identified as a key modulator of the cell cycle in glioma[39], was involved in E2f-mediated cancer progression in a model of hepatocellular carcinoma[35], and has been suggested as a potential diagnostic and prognostic marker of oral squamous cell carcinoma[40] and diffuse large B cell lymphoma[41]. Equally important, Pontin expression can predict the therapeutic outcome in lung cancer[42] and prostate tumours[43]. Besides cancer, inhibition of Pontin activity epigenetically suppresses the pro-inflammatory response of monocytes[44] and is implicated in the development of renal failure and hydrocephalus associated with ciliopathies[45]. Considering that the majority of previous findings indicated a detrimental role of Pontin expression in different cell types/organs, our present data provide an interesting phenomenon that Pontin expression in cardiomyocytes and adult heart is beneficial.

After understanding the essential role of Pontin in adult heart, it was important to also understand the mechanism. Our RNAseq data from both Pontin<sup>icKO</sup> and Pontin<sup>cTG</sup> mice pointed to a possible link between Pontin and the Hippo pathway. Subsequent analysis in cultured cardiomyocytes showed that Pontin induces YAP nuclear localization and promotes its co-transcriptional activity. The finding that the observed phenotypes in NRCM following Pontin knockdown could be rescued by expressing active YAP further confirms the mechanism that Pontin exerts its function by regulating the Hippo/YAP pathway. YAP is known to stimulate expression of genes involved in cellular growth, pro-survival and anti-apoptosis[46]. By interacting with Foxo1, YAP also induces expression of genes that protect against oxidative

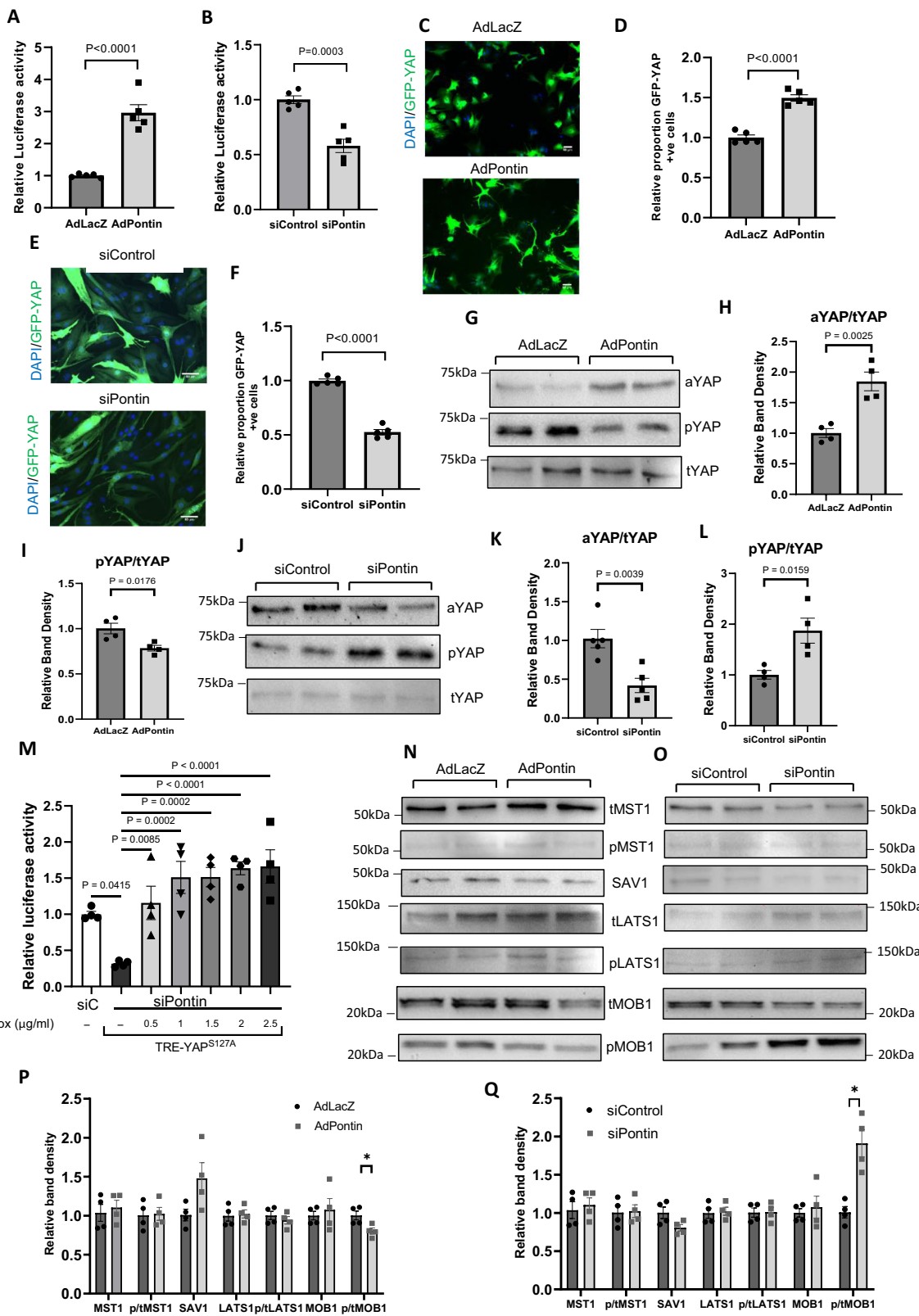

stress[47]. Therefore, Hippo/YAP modulation may explain the protective role of Pontin overexpression and the detrimental effects of Pontin knockout in adult hearts.

So, how does Pontin modulate YAP activity? Our data indicated that Pontin might regulate the core components of the Hippo pathway at both transcriptional and protein activation (phosphorylation) levels. Both Pontin overexpression and silencing in cardiomyocytes resulted in altered phosphorylation of MOB1, a key component of the Hippo pathway. Consistently, mutant Pontin[D302N], which lacks ATPase activity, failed to reduce MOB1 phosphorylation. On the other hand RNAseq data from Pontin[icKO] hearts followed by Western blot analysis indicated an upregulation of MST1 and MOB1 in knockout hearts, which if activated will induce YAP phosphorylation and sequestration/inactivation[48]. Indeed, MST1 itself is involved in inducing apoptosis via

**Fig. 7 | Pontin modulates YAP activity in cardiomyocytes. A** Analysis of NRCM overexpressing Pontin using YAP-luciferase reporter system showed enhancement of YAP activity in Pontin overexpressing cells and (**B**) reduction of YAP activity in Pontin deficient cells (*n* = 5 independent experiments in each group). YAP nuclear translocation was monitored using GFP-YAP construct in NRCM expressing Pontin (**C, D**) and NRCM lacking Pontin (**E, F**) (scale bars = 50 μm). The data suggested that Pontin expression positively modulated YAP nuclear translocation whilst Pontin knockdown reduced YAP nuclear translocation (*n* = 5 independent experiments in each group). **G** Western blot assay followed by quantification of (**H**) active and (**I**) phosphorylated YAP in NRCM overexpressing Pontin showed that Pontin over-expression increased the level of active YAP but reduced phospho-YAP expression (*n* = 5 independent experiments in each group). Consistently, Pontin knock down reduced active YAP (**J, K**) and increased phospho- YAP level (**L**) as indicated by Western blot and quantification of band density (*n* = 5 independent experiments in each group). **M** Expression of constitutively active YAP[S127A] in NRCM lacking Pontin (siPontin) rescued the reduction of YAP-luciferase signal due to Pontin knockdown (n = 4 independent experiments). **N** Representative Western blots of core components of Hippo pathway in NRCM following Pontin overexpression and (**O**) Pontin gene silencing. **P** Analysis of band density from Pontin overexpression model and (**Q**) Pontin gene silencing showed that the level of phosphorylated MOB1 was downregulated in the overexpression model and upregulated in the knockdown model, whilst expression and phosphorylation level of other members of Hippo pathway were unaltered (*n* = 4 independent experiments in each group). Data are presented as mean ± SEM. Statistical tests used: (**A–L**) two-tailed Student's *t*-test, (**M**) one-way ANOVA followed by multiple comparison test. (**P, Q**) multiple *t*-test. Source data are provided as a Source Data file.

Bcl-xL phosphorylation, which is independent of the canonical Hippo pathway[49]. Of note, the regulatory role of Pontin in modulating YAP activity is more pronounced in adult cardiomyocytes compared to neonatal myocytes, adding to the potential translational application that Pontin can be targeted to address adult cardiac diseases.

The Hippo pathway has a pivotal role both in the developmental and in adult heart. The Hippo pathway is one of the major signalling pathways that can be modulated to promote cardiac self-renewal[10,31]. Both inhibition of the upstream core components of the pathway and activation of the downstream effectors, YAP and TAZ, resulted in the induction of cardiomyocyte regeneration and survival following stress. For example, genetic inhibition of Sav1, which is one of the key upstream components of the pathway, reversed the ischaemic HF phenotype following MI in mice[10]. Consistently, overexpression of constitutively active YAP S127A induces cardiomyocyte proliferation, sufficient to reduce the scar size, as well as improve cardiac function following MI[31]. Moreover, pharmacological modulation of this pathway has also shown promising effects. For example, treatment with MST1/2 inhibitor XMU-MP-1 improves the cardiac phenotype of mice following pressure overload[32] and I/R injury[50], whereas activation of YAP/TAZ using TT-10 induces cardiomyocyte regeneration following MI[51]. Taken together, identification of regulators of the Hippo pathway that can be targeted pharmacologically may be very useful in finding new treatment strategies for heart disease.

However, it is important to note that other signalling pathways might also be modulated by Pontin. IPA analysis of Pontin[cTG] mice suggested that transcription factors such as SMAD3 and STAT6 might also be differently regulated due to Pontin overexpression, although not as significantly as TEAD1. The finding that expressions of several known YAP target genes were not significantly altered after Pontin overexpression in NRCM might be explained by possible involvement of other signalling pathways. Thus, further studies are needed to fully understand the roles of Pontin and the mechanisms by which it regulates key cellular functions.

The next question would be whether it is possible to target Pontin for therapeutic purposes. Indeed, activation of Pontin using genetic approaches, such as AAV or modRNA -mediated expression, may be a future possibility. However, pharmacologic modulation may still be the more viable approach. Several inhibitors of Pontin have been identified, for example Elkaim and colleagues have identified substances that potentially can be used as Pontin inhibitors[52]. However, based on our findings it is Pontin activation, not inhibition, that may produce beneficial effects in the heart.

Since Pontin activity depends on interaction with its paralog Reptin (Ruvbl2)[14], Pontin activation may be achieved by inhibiting Reptin activity or its interaction with Pontin. Thus, it is important to test compounds that have been shown to inhibit and/or modulate Pontin and Reptin individually as well as molecules that can modulate the Pontin-Reptin interaction complex. Indeed, more recent investigations have identified modulators of Pontin and Reptin such as sorafenib[53], CB-6644[54], and pyrazolo[1,5-*a*]pyrimidine-3-carboxamide derivatives[55] as well as a possible modulator of the Pontin-Reptin interaction[56]. Further studies are required to test the effects of these Pontin pharmacologic modulators in cardiomyocytes and in mouse models of heart disease.

In conclusion, our recent study provides comprehensive evidence on the role of a cardiac foetal gene Pontin in regulating important cellular functions in cardiomyocytes, such as cell growth and proliferation, cell survival and apoptosis. In vivo, Pontin ablation is detrimental whereas Pontin overexpression might be beneficial in protecting the heart from pathological stress. Mechanistically, Pontin may be associated with the regulation of the Hippo pathway. Overall, our findings may identify a new potential therapeutic target for protecting the heart from adverse effects following stress stimuli.

## Methods
Studies presented in this article complies comply with the relevant ethical regulations. Animal studies have been approved by University of Manchester Animal Welfare and Ethical Review Board. Human tissue samples were obtained from Asterand, who obtained informed consent for the use of human tissue samples and approval by the United Kingdom Human Tissue Authority.

### Pontin expression study
Heart tissue samples were collected from adult Sprague Dawley (SD) rats (3 months old), 2–3 days old SD rat neonates, C57Bl/6 mice (8 weeks old) following 1 day and 3 weeks transverse aortic constriction (TAC), C57Bl/6 mice following twice a day 90 min swimming for 4 weeks. Heart tissues from Rhesus monkeys with metabolic syndrome and their controls were provided by Dr Rui-Ping Xiao (Peking University, China)[20]. Human heart failure protein and RNA samples and non-heart failure controls were obtained from Asterand.

### Isolation and culture of neonatal and adult rat cardiomyocytes
Primary neonatal rat cardiomyocytes (NRCM) were derived from Sprague-Dawley rat neonates (1–3 days old). They were isolated and cultured following methods established in our laboratory[32,57]. In brief, neonatal hearts were digested in ADS solution (116 mM NaCl, 20 mM HEPES, 1 mM NaH2PO4, 5.5 mM glucose, 5.5 mM KCl 1 mM MgSO4; pH 7.35) containing 0.6 mg/ml collagenase A (Roche) and 0.6 mg/ml pancreatin (Sigma) in a shaking waterbath at 37 C for 7 min. Detached cells were collected and the digestion process was repeated for a further seven times. Cardiac fibroblasts were removed by plating on 10 cm tissue culture dishes for 1 hr to allow fibroblasts to attach. Cells in the media, which were predominantly cardiomyocytes, were collected and plated on BD Falcon Primaria plates in medium containing 68% DMEM, 17% M199, 10% horse serum, 5% FBS, 2.5 μg/ml amphotericin B and 1 μM 5-bromo-2-deoxyuridine (BrdU). Cells were incubated for 24 hr at 37 °C and after 24 h, cardiomyocytes were washed twice with PBS and maintained in a maintenance medium containing 80% DMEM and 20% M199, 1% FBS, 2.5 μg/ml amphotericin B and 1 μM BrdU before being used in experiments.

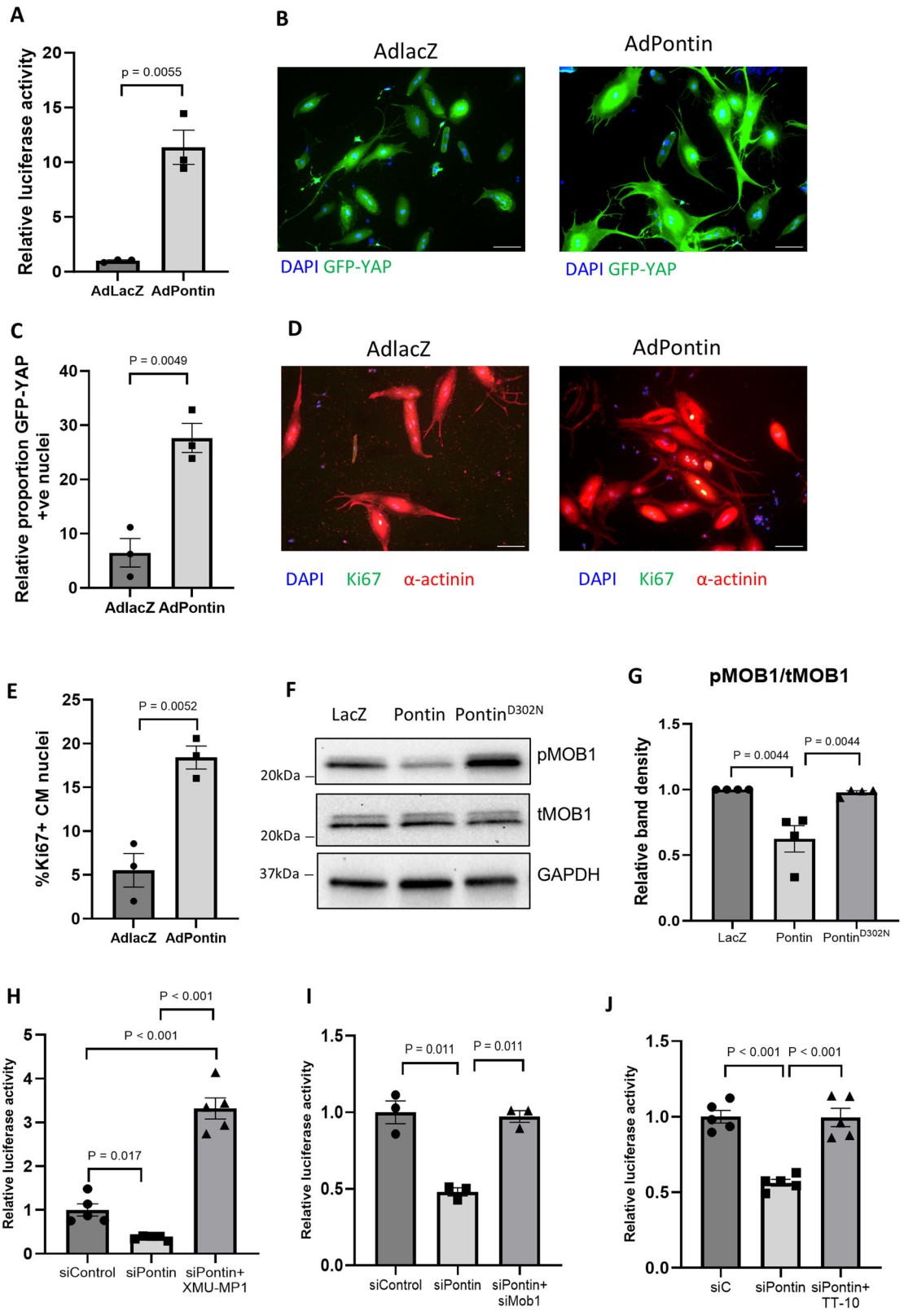

Adult rat cardiomyocytes (ARCM) were isolated from adult Sprague-Dawley rats and cultured following protocols described previously[58]. Rats were injected with sodium heparin at 1000 U/kg body weight (BW) and anaesthetized with sodium pentobarbital at 80 mg/kg BW intraperitoneally. Hearts were extracted, perfused retrogradely with $Ca^{2+}$-containing Krebs-Henseleit (KHB) buffer for 5 min and then followed by $Ca^{2+}$-free KHB buffer for further 5 min. The solution was then switched to digestion buffer ($Ca^{2+}$-free KHB buffer containing 118 U/ml collagenase type II (Gibco), 0.225 mg/ml hyaluronidase (Sigma-Aldrich) and 10 µM Blebbistatin (Cayman Chemical)) for 30 – 50 min. Following removal of aorta and atrium, hearts were minced, subjected to repeated digestion with enzyme solution (PBS containing 0.05% trypsin-EDTA) and filtered through a cell strainer (100 µm, BD Falcon). Stop buffer ($Ca^{2+}$-containing 20 mg/ml bovine

**Fig. 8 | Regulation of Hippo/YAP pathway by Pontin in adult and neonatal rat cardiomyocytes. A** Overexpression of Pontin in adult rat cardiomyocytes (ARCM) significantly increased YAP-luciferase activity ($n$ = 3 independent experiments in each group). **B**, **C** Analysis of GFP-YAP localization in ARCM showed that Pontin overexpression induced YAP nuclear translocation ($n$ = 3 independent experiments, scale bars = 100 μm). **D** Representative images of Ki67 staining in ARCM (scale bars = 100 μm) and (**E**) quantification of Ki67 positive cells suggested that Pontin overexpression significantly enhanced ARCM proliferation ($n$ = 3) independent experiments. **F** Representative Western blots and (**G**) analysis of band density showed that overexpression of Pontin significantly reduced MOB1 phosphorylation, however, overexpression of inactive mutant Pontin^D302N did not alter MOB1 phosphorylation in cardiomyocytes ($n$ = 4 independent experiments). **H** YAP activity was significantly reduced in Pontin-deficient cardiomyocytes (siRNA Pontin), however, YAP activity in these cells was restored at the level or higher than control following treatment with MST1/2 inhibitor (XMU-MP-1, 3 μM), (**I**) siRNA MOB1 and (**J**) YAP activator (TT-10, 10 μM). ($n$ = 5) independent experiments per group for (**H, J**) and $n$ = 3 independent experiments for (**I**). Data are presented as mean ± SEM. Statistical tests used: (**A–G**) two-tailed Student's *t*-test, (**G**) Kruskal Wallis test followed by multiple comparison tests, (**H–J**) and (**G**) one-way ANOVA followed by multiple comparisons test. Source data are provided as a Source Data file.

serum albumin (BSA) and 10 μM Blebbistatin) was then added to the cells. Rod-shaped cardiomyocytes were collected by gradually adding CaCl$_2$ and centrifugation at 30 x $g$ for 30 s. The cells were then resuspended in M199 (Sigma-Aldrich) supplemented with 10 mM glutathione (Sigma), 0.2 g/L BSA, 5 mM creatine (Sigma), 2 mM L-carnitine (Sigma), 5 mM taurine (Sigma), 0.1% insulin-transferrin-selenium-X (Gibco), 1% penicillin-streptomycin (Gibco), 5% FBS and 10 μM (Blebbistatin) and plated in 24 well plates on a laminin coated coverslips.

### Isolation of adult mouse cardiac fibroblasts (ACF)
ACF were isolated from 3-4 months old mice. The hearts were extracted following cervical dislocation and transferred to a tube containing ice-cold KHB buffer. Hearts were subsequently minced and digested with digestion solution (KHB buffer containing 1.5 mg/ml collagenase A (Roche) and 0.15 mg/ml protease (Sigma) at 37°C for 5 min). Minced heart tissues were then triturated several times and filtered through a cell strainer. Five millilitres FBS was added into the isolate and the digestion process was repeated at least three times. The harvested fibroblasts were pooled and centrifuged for 5 min at 220 x $g$. The pellets were then resuspended in ACF media (DMEM (Gibco) supplemented with 15% FBS, 1% penicillin streptomycin and 1% non-essential amino acids (Gibco)). The following day, the media was replaced with ACF media containing 10% FBS.

### Pontin overexpression and gene silencing in NRCM and ARCM
We generated adenovirus to overexpress Pontin in NRCM and ARCM. Plasmid containing human Pontin (*RUVBL1*) cDNA was purchased from Origene. A 3xFlag sequence was inserted at the N-terminus of the cDNA before cloning to pENTR11 shuttle vector (Invitrogen). We then used the Gateway adenovirus system (Invitrogen) to generate adenovirus plasmid carrying the flag-tagged Pontin cDNA. Adenovirus particles were produced using standard methods as described in our previous publication[22].

Pontin^D302N mutant lacking ATP-ase activity was generated by replacing the aspartic acid residue at the position 302 with asparagine using QuikChange Site-Directed Mutagenesis Kit (Agilent Technologies). Forward (5′-GGTGTGCTGTTTGTTAATGAGGTCCACATGCT-3′) and reverse (5′-ATGTGGACCTCATTAACAAACAGCACACC-3′) primers were used to generate the mutant Pontin^D302N construct from the wild type Pontin cDNA. Adenovirus particles were produced using standard methods as described above.

We used siRNA targeting Pontin (Sigma, #SASI Rn02-00268862) or MOB1 (Sigma # Sasi_RnO1_00116665) to knockdown Pontin or MOB1 in NRCM. Pontin, MOB1 or control siRNA were transfected to NRCM using Dharmafect transfection reagent (Dharmacon) following the protocol recommended by the manufacturer.

### Cell proliferation assays
The rate of NRCM proliferation was assessed by detecting the expression of cell cycle marker Ki-67 and mitosis marker pHH3 as well as the rate of EdU incorporation. NRCM -overexpressing or -lacking Pontin together with control cells were fixed with 4% paraformaldehyde for 15 min, permeabilised with 0.1% Triton X-100 in PBS for 10 min

and blocked with 0.5% BSA for 1 h at RT before immunofluorescence detection. Primary antibodies used include rabbit anti Ki-67 antibody (Abcam), Alexa Fluor 488-conjugated rabbit anti pHH3 antibody (Cell Signaling Technology), and mouse anti α-actinin antibody (Sigma). Secondary antibodies were FITC-conjugated anti rabbit and Alexa-Fluor 647-conjugated anti mouse antibody (Jackson Immunoresearch). Cells were then counterstained using DAPI. We used ClickIT EdU Imaging Kit Alexa Fluor-488 (Life Technologies) for EdU incorporation analysis following methods recommended by the manufacturer.

### Cellular hypertrophy analysis
NRCM -overexpressing or -lacking Pontin and control cells were treated with 1 μM Angiotensin II (Sigma) for 48 h. Then, NRCM were stained with anti-α-actinin antibody (Sigma) and the cardiomyocyte circumference was measured using ImageJ software. Brain natriuretic peptide (BNP) promoter activation was monitored using a BNP-luciferase adenoviral reporter system as described in previous publication[19].

### Analysis of cardiomyocyte apoptosis and survival in vitro
We performed TUNEL analysis to determine cardiomyocyte apoptosis. Following Pontin overexpression or knockdown, NRCM were treated with Angiotensin II (1 μM, 48 h). We used In situ Cell Death Detection Kit (Roche) to detect TUNEL positive nuclei following the protocol recommended by the manufacturer. For in vitro cell survival assay, we treated NRCM with 200 μM hydrogen peroxide (Sigma) for 4 h to induce oxidative stress. Following this, the media was replaced with maintenance media containing AlamarBlue reagent (Thermo Fisher) at 1:10 dilution. After 4 h, the media were transferred into a 96 well-plate and fluorescence signal was detected using a spectrofluorophotometer (FLUOStar Omega, BMG Labtech) at 540–570 nm (excitation) and 580–610 nm (emission).

### Luciferase assays
For signalling pathway screen analysis we used our adenovirus based luciferase reporter constructs to monitor the activation of AP1, STAT3, YAP (GAL4-TEAD), Wnt, NFκB and NFAT promoter elements in the H9c2 cardiomyoblast cell line. The construction of the luciferase reporters were described in our previous publication[57].

H9c2 cells were obtained from ATCC. They were maintained in DMEM with 10% FBS, 1% non-essential amino acids solution (Gibco) and 100 units/ml penicillin and streptomycin. H9c2 cells were seeded in 48-well plates at $2 \times 10^5$ cells per well, along with adenovirus carrying either AP1, STAT3, YAP (GAL4-TEAD), Wnt, NFκB or NFAT luciferase constructs. After 24 h the media was changed and cells were treated with adenovirus overexpressing Pontin or transfected with siRNA to knockdown Pontin. After 48 h luciferase activity was assessed using a Lumat LB9507 Tube Luminometer (Berthold) and Luciferase detection reagent (Promega).

YAP-luciferase activity was also measured in NRCM. For this purpose NRCM were plated onto 24 well plates and then treated with either adenovirus overexpressing Pontin or siRNA to knockdown Pontin. After 24 h for the overexpression system or 48 h for the gene

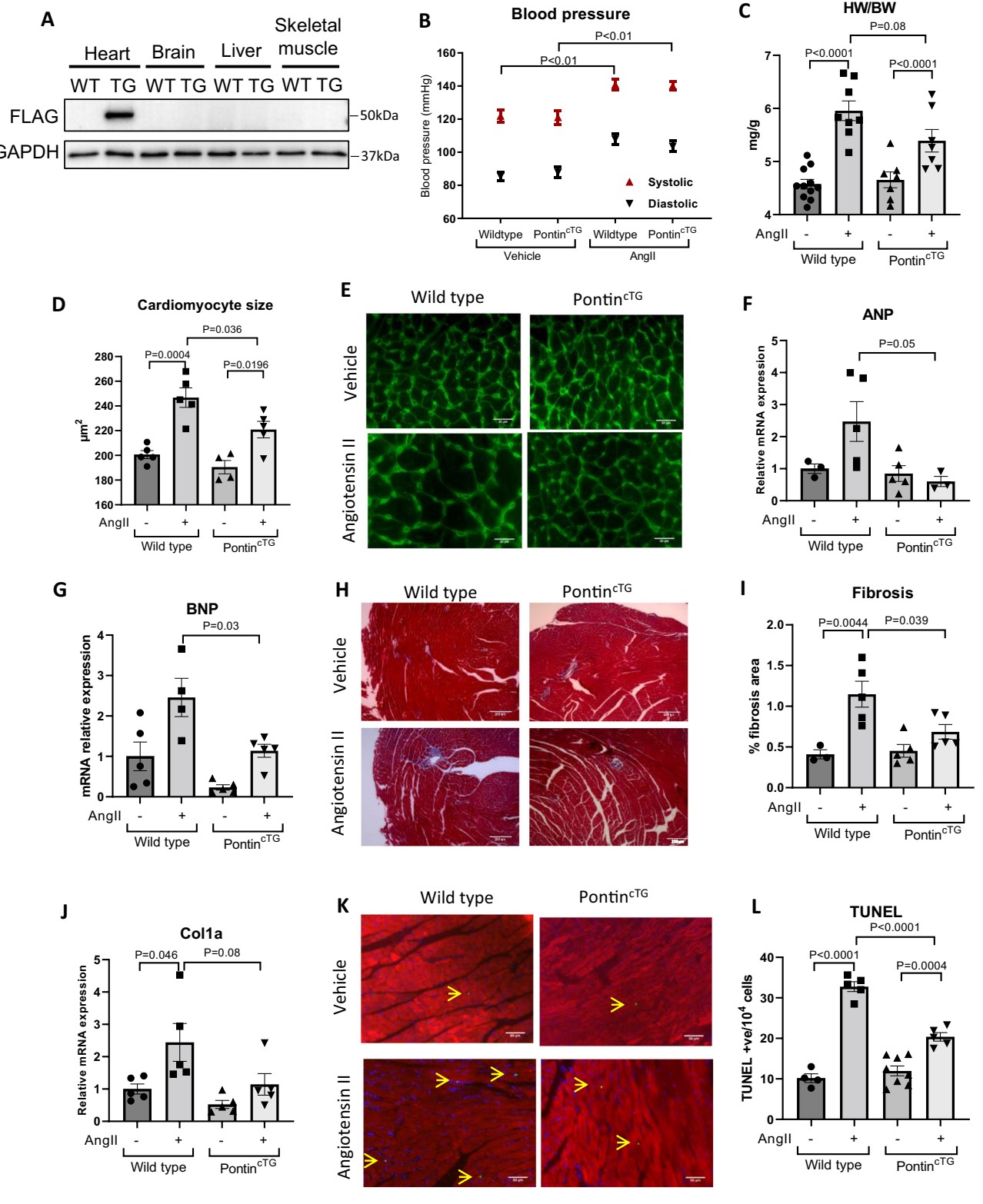

TUNEL, DAPI, α-actinin

knockdown system, cells were lysed, and the luciferase signal was measured as described above.

## YAP nuclear translocation

To analyse YAP nuclear translocation, we used an adenovirus based GFP-YAP construct. The generation of GFP-YAP adenovirus was described in our previous publication[32]. NRCM overexpressing or lacking Pontin were seeded onto laminin coated coverslips in 24 well plates. GFP signals were then detected using fluorescent microscopy.

## YAP rescue experiments

We used YAP$^{S127A}$ inducible expression system to rescue the phenotypes due to Pontin knockdown in NRCM. Adenoviruses containing TRE-YAP$^{S127A}$ and CMV-rtTA constructs were generated in our

**Fig. 9 | Trangenic mice with cardiomyocyte specific overexpression of Pontin protected against AngII induced pathological remodelling. A** Western blot images showing cardiac specfic expression of Flag tagged Pontin transgene in cardiomyocyte specific Pontin transgenic (Pontin^cTG) mice. **B** Assessment of blood pressure of WT and Pontin^cTG mice after two weeks of Vehicle or AngII treatment. In both groups of mice, AngII significantly increased blood pressure by 20–25%, however no difference was observed between WT and Pontin^cTG (WT, *n* = 12 mice, Pontin^cTG, *n* = 7 mice, WT+AngII, *n* = 8 mice, Pontin^cTG+AngII, *n* = 7 mice). **C** Assessments of cardiac size (HW/BW ratio, WT, *n* = 11 mice, Pontin^cTG, *n* = 8 mice, WT+AngII, *n* = 7 mice, Pontin^cTG+AngII, *n* = 7 mice), **D, E** cardiomyocyte cross- sectional area (WT, *n* = 5 mice, Pontin^cTG, *n* = 4 mice, WT+AngII, *n* = 4 mice, Pontin^cTG+AngII, *n* = 4 mice)(scale bars = 20 μm), (**F**) hypertrophic markers ANP and

(**G**) BNP (WT, *n* = 5 mice, Pontin^cTG, *n* = 4 mice, WT+AngII, *n* = 4 mice, Pontin^cTG+AngII, *n* = 4 mice) revealed a reduction in hypertrophic response in Pontin^cTG mice. **H** Representative Masson's trichrome stained heart sections (scale bars = 200 μm), (**I**) quantification of fibrotic area (WT, *n* = 3 mice, Pontin^cTG, *n* = 5 mice, WT+AngII, *n* = 5 mice, Pontin^cTG+AngII, *n* = 5 mice) and (**J**) measurement of collagen I expression (*n* = 5 in each group) showed a significant reduction in fibrosis in Pontin^cTG mice. **K, L** Apoptosis level was also reduced in Pontin^cTG mice following Ang II stimulation as shown by TUNEL assay (scale bars=50 μm)(WT, *n* = 4 mice, Pontin^cTG, *n* = 8 mice, WT+AngII, *n* = 5 mice, Pontin^cTG+AngII, *n* = 5 mice). Data are presented as mean ± SEM. Statistical tests used: (**B**–**L**) one-way ANOVA followed by multiple comparisons test. Source data are provided as a Source Data file.

laboratory previously[26]. YAP^S127A expression was induced by treatment with doxycycline (0.5–2.5 μg/ml).

### Western blots
Total protein from NRCM or heart tissues was extracted via lysis in RIPA buffer containing 1% IGEPAL CA-630, 0.5% sodium deoxycholate, 0.1% SDS, 0.5 mM phenylmethylsulphonyl fluoride, 500 ng/ml Leupeptin, 1 mg/ml Aprotinin and 2.5 mg/ml Pepstatin A. For Western blot analysis, protein was separated using sodium dodecyl sulfate polyacrylamide gel electrophoresis (SDS-PAGE), transferred to Immobilon-polyvinylidene difluoride membrane (Millipore) and blocked in 1–5% BSA or non-fat milk before hybridization with respective primary and secondary antibodies. A list of antibodies used is available in Supplementary Table 2.

### Quantitative RT-PCR
RNA was isolated from cultured NRCM or heart tissues using Trizol reagent (Invitrogen) following the manufacturer's recommended protocol. The QuantiTect-SYBR Green RT–PCR kit (Qiagen) was used for qRT–PCR analysis. mRNA expression levels were determined using the ΔΔCt method and are presented as fold induction of target gene transcripts relative to control group with GAPDH as reference gene. qPCR reactions were performed in an ABI 7500 Fast System (Applied Biosystems). A list of primers are described in Supplementary Table 3.

### Generation of Pontin inducible cardiac specific knockout mice
Animal studies were conducted in accordance with the United Kingdom Animals (Scientific Procedures) Act 1986. Institutional approval was obtained from the University of Manchester Animal Welfare and Ethical Review Board. Mice were maintained in the University of Manchester BSF Facility in a standard housing condition for laboratory animals. Mice were maintained on a 12 h light/dark cycle in a controlled temperature of 19–22 °C and humidity of 40–65%. Mice were fed with standard chow diet (BK001, Special Diet Services, UK).

Pontin gene-targeted mice were obtained from the European Conditional Mouse Mutagenesis Programme (EUCOMM). The targeted allele was generated using a "knock-out first" strategy (PMID 21677750). The exon 3 of mouse Ruvbl1 gene (Pontin) was targeted using a construct carrying FRT and loxP sites as described in Fig. S3. Heterozygous mice carrying the conditional targeted allele were crossed with CAG-Flpo transgenic mice to remove the lacZ and neo markers resulting in clean Pontin^floxed (Pon^flox) mice. To generate a cardiomyocyte specific inducible KO we crossed homozygous Pon^flox mice with αMHC-MerCreMer transgenic mice (αMCM) generating αMCM- Pon^flox mice. All mice were bred on a C57Bl/6 background.

Pontin inducible cardiomyocyte knockout (Pontin^icKO) were obtained by injecting 8–10 weeks old male αMCM- Pon^flox mice with 40 mg/kg BW of tamoxifen. Non tamoxifen-injected αMCM- Pon^flox and αMCM mice (tamoxifen and non-tamoxifen treated) were used as controls. A list of genotyping primers are described in Supplementary Table 4.

### Generation of transgenic mice with cardiac specific over-expression of Pontin
A flag-tagged human Pontin cDNA was cloned downstream of the mouse α-myosin heavy chain (α-MHC) promoter. The linearised construct was then injected into a single cell mouse embryo by standard pro-nuclear injection technique. Transgenic founders were identified by PCR and were bred on a C57Bl/6 background. Non-transgenic littermates were used as controls.

### Angiotensin-induced hypertrophy model
Ten weeks old male *αMHC-MerCreMer* (αMCM) and *αMHC-MerCreMer Pontin^flox/flox* (αMCM-Pon^flox) mice were injected with tamoxifen intraperitoneally at a dose of 40 μg/g BW. After 7 days, Angiotensin II (1.5 mg/kg BW/day) or saline were infused via mini-osmotic pumps (Alzet). Blood pressure was monitored using a tail-cuff CODA volume pressure recording (VPR) system (CODA, Kent Scientific, Torrington, USA). Experiments were terminated at day 3 after mini-pump implantation. Echocardiography analysis was conducted before termination. Heart tissues were collected for histological and molecular examinations.

Ten weeks old male Pontin^cTG mice and their WT littermates were subjected to Ang II treatment at a dose of 1.5 mg/kg BW/day using mini osmotic pump for 14 days. Blood pressure was monitored as above. After 14 days of treatment echocardiography analysis was performed and the mice were sacrificed. Heart tissues were collected for histological and molecular examinations.

### Echocardiography
To assess cardiac morphology and function of Pontin^icKO and Pontin^cTG mice at basal conditions and following Ang II treatment, transthoracic echocardiography was conducted using a Visualsonics Vevo 770 imaging system fitted with a 14 MHz transducer. Mice were anaesthetised using 1.5% isofluorane, and images were acquired in parasternal long and short axis views to measure left ventricular internal diameter (LVID) at diastole and systole, posterior wall (dPW), and interventricular septal (dIVS) thicknesses in systole and diastole. Ejection fraction was then calculated from these measurements.

### Histology analysis
For histological analysis, heart tissues were fixed in 4% paraformaldehyde in PBS for 24 h at 4 °C. Then, the tissues were processed overnight using a Leica automated tissue processor and embedded in paraffin wax prior to sectioning at 5 μm thickness using an automated rotary Leica RM2255 microtome. Haematoxylin and eosin staining was carried out to assess cardiomyocyte size. Masson's trichrome staining was conducted to evaluate the level of fibrosis. Cardiac tissue sections were also stained using TUNEL reagents (In situ Cell Death Detection Kit, Roche) for detection of apoptosis. Dihydroethidium (DHE) staining was performed to analyse the level of reactive oxygen species (ROS) production in the tissue. In brief, sections were dewaxed and dehydrated and then incubated with DHE (Invitrogen, 1:2000 dilution in PBS from 5 mM stock) for 30 min at 37 °C with coverslips attached. After incubation, images were immediately taken by Zeiss™ fluorescence

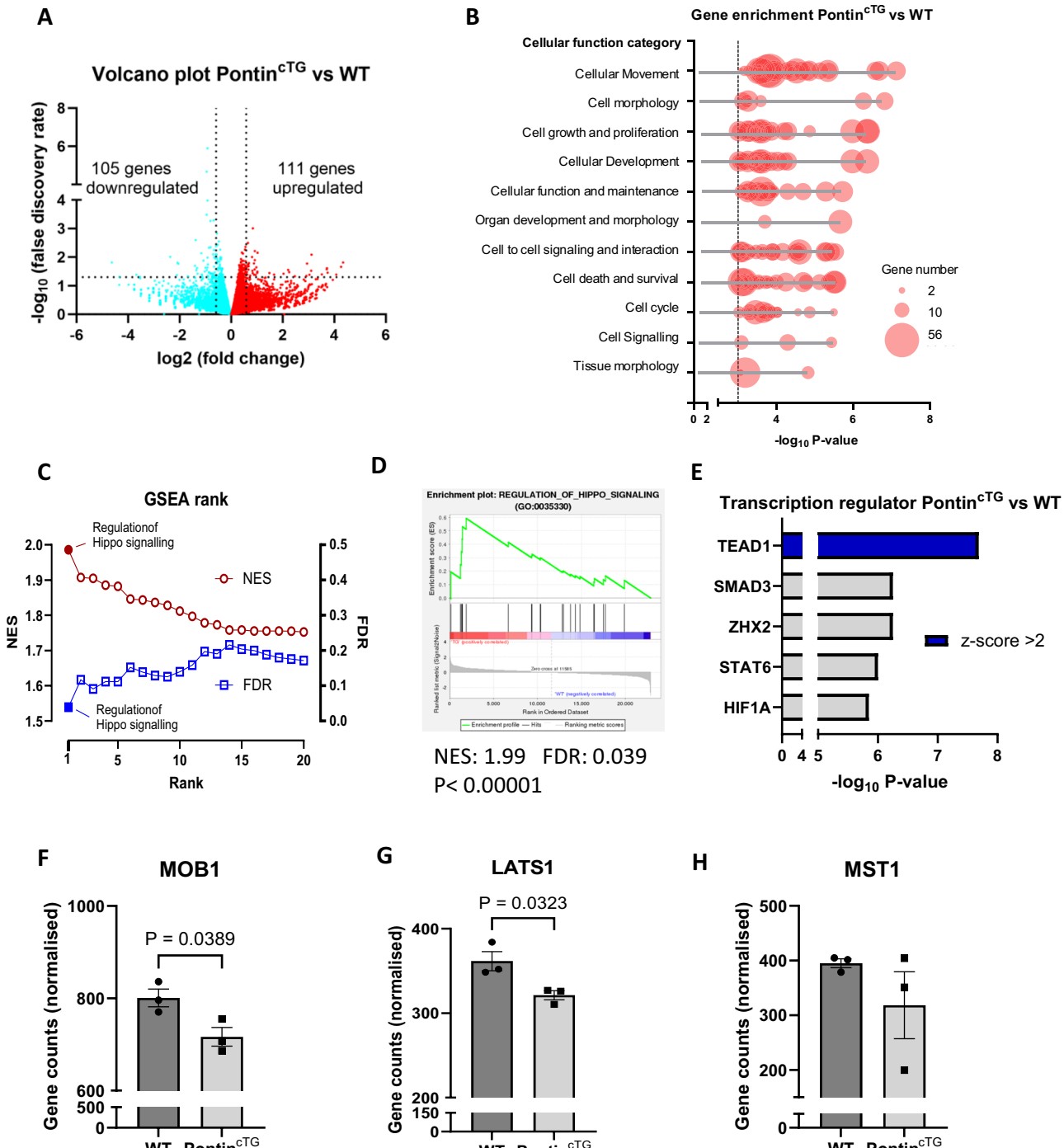

**Fig. 10 | Transcriptome analysis of PontincTG mice following Angiotensin II infusion indicated that Pontin might regulate Hippo pathway in the transgenic mouse model. A** Volcano plot showing the number of differentially expressed genes (DEGs) in heart tissues of PontincTG mice compared to WT littermates following stimulation with Ang-II (1.5 mg/kg BW/day for 2 weeks ($n = 3$ mice in each group)). **B** Ingenuity pathways analysis (IPA) DEGs showed enrichment of sets of genes related to specific cellular functions. The negative log of *P*-values are shown in *X*-axis. The size of the dots represents the number of DEGs in each category. **C** GSEA analysis indicated that the top-ranked gene set that was differently expressed in PontincTG mice was "Regulation of Hippo Signalling"

based on net enrichment score (NES) and FDR values. **D** Enrichment plot from GSEA analysis showing the NES and FDR values of gene set "Regulation of Hippo Signalling". **E** IPA analysis identified TEAD1 as the top transcription factor that was differently regulated in PontincTG hearts following Ang-II stimulation. Analysis of individual genes revealed that the expressions of (**F**) MOB1 and (**G**) LATS1 were significantly downregulated in PontincTG mice, whereas expression of (**H**) MST1 showed trend of reduction although it did not reach significance. ($n = 3$ mice in each group). Data are presented as mean ± SEM. Statistical tests used: (**F**–**H**) two-tailed Student's *t*-test. Source data are provided as a Source Data file.

microscope (Carl Zeiss, Jena, Germany) then subsequently analysed using the integrated density function in ImageJ software.

## Immunohistochemistry staining

Tissue sections (5 μm) were deparaffinised on a heat block at 80 °C for 2 min and cleared in histoclear for 10 min. Sections were then sequentially rehydrated in decreasing percentages of industrial methylated spirit in water (100%, 90%, 75%) and washed in distilled water. Sections were then permeabilised in 0.1% Triton X-100 for 8 min and antigen retrieval was performed by incubating sections in 10 mM sodium citrate, pH 6.0 in a 90 °C waterbath for 20 min. YAP staining was then conducted using a HRP/DAB detection IHC kit (Abcam #ab64264) following manufacturer instructions. Primary antibodies used were anti-phosphoYAP (CST #13008), anti-YAP (Santa Cruz Biotech #sc-376830). Sections were imaged on a 3D Histech Panoramic 250 Flash II slide scanner and 5 40x magnification images of each mouse heart sample were analysed for DAB intensity using ImageJ.

## RNAseq analysis

Heart tissues from Pontin[icKO] mice (αMCM-Pon[flox] + TAM) and control (αMCM + TAM) were collected at 1 week and 3 week following tamoxifen injection. For the transgenic overexpression model, heart tissues from Pontin[cTG] mice and WT littermates were collected after infusion with Ang-II for 2 weeks (1.5 mg/kg BW/day via mini-osmotic pump). Sequentially, RNA was extracted using Trizol reagent. Sequencing library for RNA sequencing was prepared with TruSeq RNA Library Prep Kit v2 (Illumina), according to the manufacturer's protocols. RNA sequencing was performed using the Illumina HiSeq2500 sequencer with 100 bp paired-end reads. The quality of the reads was assessed using FastQC Version 0.10.1 and filtered by using Trimmomatic v0.39. The reads were then mapped on the reference mm10 with Gencode M25 gtf (version 2.7.7a) using STAR. For gene-level analysis, raw counts were produced using STAR using Gencode M25 gtf in GENE. Differential expression analysis was conducted using with DESeq2 (v1.32.0 R4.1.1) from Bioconductor (https://www.bioconductor.org/). An outlier identified by principal component analysis was excluded from downstream analysis. Differentially expressed genes between samples from *Pontin[icKO]* mice and control mice at 2 time points (1 week and 3 week) were identified using DESeq2. Gene lists were annotated based on biomaRt version 2.48.3.

Identification of cellular function that were enriched in Pontin[icKO] were performed using Ingenuity Pathway Analysis (IPA, Qiagen). Signalling pathway analysis was performed by using GeneSet enrichment analysis (GSEA) as previously described[59].

## Statistics and reproducibility

For the in vivo and in vitro analysis, we used one way ANOVA followed by multiple comparisons test (Tukey) to compare means in multiple group experiments. In the case of two groups experiments Student's *t*-test was used. For data that were not distributed normally we used non-parametric tests (Kruskall-Wallis test followed by Dunn's multiple comparison). All of the data are presented as mean ± SEM. We used GraphPad Prism software ver.10.2.2 (GraphPad Software, LLC) for statistical analysis and data presentation. A *P*-value <0.05 (two tailed) was considered statistically significant.

For the in vivo experiments, mice were randomly selected to be included in each experimental group, e.g. treatment with AngII to induce hypertrophy or with tamoxifen to delete Pontin. For in vitro experiments, cells in each cultured TC well were randomly selected for treatment (e.g. overexpression or gene silencing of Pontin). The investigators were blinded in analysing the in vivo data, such as echocardiography. For in vitro experiments, there was no blinding process since the same researcher performed both treatment and phenotype analysis. For the in vivo experiments, sample size was determined based on Pilot data using Pontin transgenic mice. Sample size calculations at a

power of 80% with alpha of 0.05 (two sided *t*-test) suggest that a minimum of *n* = 5 mice per group are required. This sample size calculation was used for the analysis of cardiac phenotype of Pontin knockout and transgenic mice. No data were excluded from the analysis.

## Reporting summary

Further information on research design is available in the Nature Portfolio Reporting Summary linked to this article.

## Data availability

All of the data from this study are presented in the 'Results' section and Supplemental Materials of this paper. The data generated in this study are provided in Source Data file. RNA seq data for Pontin[icKO] experiment have been deposited in the ArrayExpress database under accession number E-MTAB-14659. RNA seq data for Pontin[cTG] experiment have been deposited in NCBI's Gene Expression Omnibus under accession number GSE282958. Source data are provided with this paper.

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

## Acknowledgements

We thank Dr. Rui-Ping Xiao (Peking University, China) for providing heart tissue samples from Rhesus monkeys. We thank Leo Zeef and Andy Hayes of the Bioinformatics and Genomic Technologies Core Facilities at the University of Manchester for providing support with regard to RNA-seq. We thank the University of Manchester Biological Services Facility and the Bioimaging Facilities for technical support. This study was supported by British Heart Foundation Programme Grant (RG/F/21/110055 to D.O). B.L. was supported by an Indonesian LPDP (Lembaga Pengelola Dana Pendidikan/Indonesia Endowment Funds for Education) PhD scholarship.

## Author contributions

D.O. conceived the scientific ideas, oversaw the project, designed in vivo and in vitro experiments, analysed data and wrote the manuscript. B.L. and A.B.N. designed and conducted in vivo and in vitro experiments, analysed data, performed bioinformatics analysis and wrote manuscript. T.A.B. designed and performed in vitro analysis and in vivo studies on Pontin transgenic mice. B.N., S.P. and M.Z. performed in vivo surgical experiments. N.S. performed in vitro experiments and edited manuscript. R.P., E.T. and F.M.B. performed in vitro experiments. A.D'S and E.J.C. supervised and helped designing in vivo experiments. X.W. provided reagents and helped data interpretation.

## Competing interests

The authors declare no competing interests.
