## [Peer Review File · Nature Communications]

REVIEWER COMMENTS

Reviewer #1 (Remarks to the Author):

The authors identified Pontin as a novel modulator of adverse cardiac remodelling. By generating an inducible cardiomyocyte specific knockout (icKO) mouse model, the authors examined the function of Pontin in cardiac remodeling. Their analysis, involving RNA-seq, unveiled the activation of the Hippo pathway in the Pontin icKO mice. However, there are some major concerns that should be addressed to strengthen the conclusion.

Major points

1. The primary objective of this study was to shed light on the role of Pontin in cardiomyocytes, especially in the context of cardiomyopathy. However, a comprehensive molecular mechanism detailing how Pontin participates in the regulation of cardiomyopathy is conspicuously absent. Notably, the study observed differential Pontin localization between cardiomyocytes and cardiac fibroblasts, but the mechanisms underlying this discrepancy remain unexplored. It is imperative for the authors to investigate whether Pontin's localization has an impact on the cardiac remodeling process.
2. It was assumed that the MCM promoter specifically expresses Cre in cardiomyocytes. It is essential to validate the cardiomyocyte-specific knockout of Pontin. Although this should be demonstrated by immunoblot and qRT-PCR analyses to compare the expression levels of Pontin between cardiomyocytes and cardiac myoblasts, the authors showed stronger expression of Pontin in cardiac myoblasts than in cardiomyocytes. Furthermore, cardiac myoblasts were found to express Pontin in both the nucleus and cytosol, unlike cardiomyocytes which express Pontin only in the cytosol (Fig 1G). This suggests that Pontin might function in cardiac myoblasts, potentially playing a role in maintaining cardiac homeostasis. Therefore, it is essential to exclude possible effects of knockout of Pontin in cardiac fibroblasts.
3. According to Fig. 1A, the expression of Pontin is high in neonatal hearts, but it decreases in adult hearts. However, many experiments were conducted by overexpressing Pontin in neonatal rat cardiomyocytes (NRCM), where Pontin was expected to be highly expressed, or by inducing knockout of Pontin in adult mice, where Pontin was expected to be expressed at a lower level. The authors need to check whether overexpression of Pontin in adult rat cardiomyocytes leads to more pronounced phenotypes.
4. The authors utilized rat, human, and Rhesus monkey samples in Fig1. However, for the mechanistic study, the authors exclusively used a rat cell line. It raises the question of whether the mechanism by which Pontin expression induces YAP activity is applicable to other species. The authors should utilize Pontin conditional TG mice samples for immunoblot analysis to measure p-MOB1 levels and for immunohistochemistry (IHC) to evaluate YAP, p-YAP, etc.
5. A mouse model overexpressing Flag-tagged Pontin specifically in cardiomyocytes was generated to demonstrate the protective function of Pontin against AngII-induced pathological cardiac

remodeling. It would have been beneficial to utilize this mouse model for further in vivo investigation of Pontin's function. Given their observation of reduced protein expression of Pontin under cardiac stress in Figs. 1B, 1C, and 1H, it would be more reasonable to present the phenotype (Fig. 8) and the altered Hippo pathway to elucidate how Pontin protects against pathological remodeling through Hippo pathway modulation. Additionally, performing RNA-seq and validating the altered Hippo pathway in the Pontin overexpressing mouse model could complement the results obtained from the icKO mouse model.

6. While Pontin expression was observed to decrease in pathological conditions across various mammalian heart tissues (Fig. 1B-D), it is important to present an appropriate marker that represents each pathological condition. Additionally, it is recommended to include a marker for phenylephrine treatment in NRCM (Fig. 1H) to ensure proper treatment.

7. Following the RNA-seq data acquired from Pontin icKO mice, the authors confirmed the upregulation of specific Hippo target genes through qRT-PCR analysis. To provide additional confirmation of Hippo pathway activation, it would be beneficial for the authors to perform active YAP staining on mouse heart tissues. This complementary approach would strengthen the evidence for Hippo pathway activation and enhance the comprehensiveness of their study.

8. It is important to elucidate the underlying molecular mechanism that connects p-MOB1 and Pontin, whether they have a direct relationship or if there is an intermediary mediator bridging their interaction. Further investigation is required to uncover the intricacies of this molecular pathway and to establish the causal relationship between these components.

9. In Figures 7M and S7B, the authors observed a decrease in the phosphorylation level of MOB1 in Pontin overexpressing cells, but no discernible difference in the expression of YAP target genes. To bolster their findings, the authors should undertake further validation of the molecular mechanisms underpinning this observation with additional experiments or analyses.

10. To strengthen their findings in Figures 2 and 3, the authors should conduct rescue experiments by reintroducing Pontin into the knockdown cell lines. This would help further elucidate the role of Pontin and provide additional insights into the mechanisms in their study.

Minor points

1. It is crucial to confirm whether the same image was utilized for both MOB1 and YAP immunoblot data in Figure 6F.

2. It is essential to present the immunofluorescence data in a manner that includes not only the merged view but also individual channels displayed separately.

3. In Figure 7C (upper) and Figure 7E (upper), both images serve as control references, and therefore, they should exhibit similar characteristics. However, it is noteworthy that in Figure 7E (upper), the localization of YAP within the nucleus appears to be less distinct when compared to Figure 7C (upper)

Reviewer #2 (Remarks to the Author):

The authors show that endogenous Pontin expression in cardiomyocytes plays a protective effect in the heart and that the protective effect of Pontin may be mediated through activation of YAP.

General:

Although the authors suggest that Hippo inactivation may mediate the detrimental effect of Pontin downregulation in cardiomyocytes in the mouse model, the results are only correlative, and the involvement of other mechanisms cannot be excluded.

Although in vitro experiments with neonatal myocytes may show that Pontin has cell-autonomous effects in cardiomyocytes, the quality of the data is generally modest.

Specific:

In Figure 1, the authors should isolate cardiomyocytes and non-myocytes from the mouse heart and show the level of pontin in cardiomyocytes and non-myocytes separately. The data shown in Fig. 1G is unclear and does not allow the comparison of the protein levels among various cell types.

In Figures 2 and 3, the quality of the experiments is modest. The quality of the cell staining is modest and there are many non-myocytes in the culture. The Y axis in some graphs is not presented in a full scale in Fig. 2. Thus, the effect of pontin modulation is modest.

In Figure 4, the authors should show cell type-specific downregulation of Pontin in cKO mice.

In Figures 5E and 6, the authors should show the level of phosphorylation of Mst1 and YAP in cardiomyocytes.

Although the study suggests that cardiomyocyte-specific downregulation of Pontin may stimulate the Hippo pathway. How it affects the activity of YAP in the cKO heart is not clearly shown. Again, the activity of YAP should be assessed in a cardiomyocyte-specific manner.

In Figure 8, the authors should show the level of pontin overexpression in cardiomyocytes in PontincTG. The immunoblot analyses should be conducted with anti-Pontin antibody, not anti-Flag. It would also be nice if the authors could show to what extent Pontin overexpression in PontincTG mice rescues the level of Pontin expression in cardiomyocyte during stress.

Minor:

The authors should have addressed as to how ponin downregulation leads to the activation of the Hippo pathway and inhibition of YAP.

Reviewer #3 (Remarks to the Author):

The manuscript submitted by Dr. Oceandy and colleagues presents interesting observations related to the Pontin gene in adult cardiac physiology. In order to study the role of Pontin in the mouse heart, the authors generated both an inducible cardiac-specific knock-out and a cardiomyocyte-specific overexpression of Pontin. Knock-out of Pontin generated hypertrophic phenotypes and exacerbated Angiotensin-II treatment, while overexpression had minimal baseline effects, but ameliorated Angiotensin-II treatment. Between cell culture systems and the in vivo models, the authors identify Pontin may be regulating core Hippo signaling components (both abundance and post-translational modifications) with Pontin levels positively correlating with YAP activity.

Specific comments:

- In Figure 4, it is described that the phenotype emerges between the 3rd and 4th week after tamoxifen. What happens after 4 weeks post-tamoxifen? Do the animals continue to get worse or does the phenotype stabilize at ~40% EF? It seems like 4 weeks was a premature endpoint.
- In the text related to the RNA-seq (line 210), it states that the cut-off was P-value < 0.05 and log₂ fold change >1 or <-1, but in Figures 5A and 5B, the Y-axis and dotted line appear to be in relation to

the false discovery rate. Please confirm the method of cut-off, and if it was a P-value cut-off, please justify not using the multiple testing correction.

- In Figure 5, is there any connection between the genes observed at 1 week post-tamoxifen and the genes observed at 3 weeks post-tamoxifen? Was the ~400 differentially expressed genes nested within the ~2000 differentially expressed genes?

- In Figures 6-7, the authors are suggesting that Pontin acts upstream of Hippo signaling to regulate YAP activity; however, there are some oddities that are making it unclear about the relationships between core members of the Hippo cascade. For example, there are increased levels of total MST1 and total MOB-1 and phosphor-MOB-1, but not downstream components like LATS1/2. Using the NRCMs, can the authors further flesh out the interaction between Pontin, Hippo signaling pathway, and YAP? Specifically, can Pontin knockdown phenotypes (Figures 2&3) be rescued by MOB1 knockdown? Can Pontin knockdown phenotypes be rescued by LATS1/LATS2 knockdown? Can Pontin knockdown be rescued with constitutively active YAP? Can the effects of Pontin overexpression be reversed by YAP knockdown (is this direct YAP activity)?

POINT BY POINT RESPONSE TO REVIEWERS' COMMENTS

Reviewer #1 (Remarks to the Author):

The authors identified Pontin as a novel modulator of adverse cardiac remodelling. By generating an inducible cardiomyocyte specific knockout (icKO) mouse model, the authors examined the function of Pontin in cardiac remodeling. Their analysis, involving RNA-seq, unveiled the activation of the Hippo pathway in the Pontin icKO mice. However, there are some major concerns that should be addressed to strengthen the conclusion.

Major points

1. The primary objective of this study was to shed light on the role of Pontin in cardiomyocytes, especially in the context of cardiomyopathy. However, a comprehensive molecular mechanism detailing how Pontin participates in the regulation of cardiomyopathy is conspicuously absent. Notably, the study observed differential Pontin localization between cardiomyocytes and cardiac fibroblasts, but the mechanisms underlying this discrepancy remain unexplored. It is imperative for the authors to investigate whether Pontin's localization has an impact on the cardiac remodeling process.

We thank the reviewer for raising this important issue. Our data showed that Pontin is indeed localised in both the cytoplasm and nucleus, and in cardiac fibroblasts it is predominantly located in the nucleus (Fig. 1H); however, it is not known whether Pontin sub-cellular localization is associated with the cardiomyocyte remodelling process. As an initial step to address this question, we performed experiments to analyse Pontin localization in neonatal rat cardiomyocytes and cardiac fibroblasts following treatment with angiotensin II (Ang II). Ang II treatment was chosen since this would be relevant to the pathological models applied in our cKO and TG mice. Immunofluorescence analysis, as shown in figure 1J indicated that there was no marked change in the subcellular location of Pontin following angiotensin II treatments in both cardiomyocytes and cardiofibroblasts compared to control. This suggests that Pontin might not be translocated from the cytoplasm to the nucleus (or vice versa) in the Ang-II-induced cardiomyocyte/cardiofibroblast remodelling; however, we agree that more experiments need to be done to further understand Pontin sub-cellular localization in other pathological models.

We have modified our manuscript to address this issue as follows:

- Addition of new figure 1J*
- Modification of text in line 112-113*

2. It was assumed that the MCM promoter specifically expresses Cre in cardiomyocytes. It is essential to validate the cardiomyocyte-specific knockout of Pontin. Although this should be demonstrated by immunoblot and qRT-PCR

analyses to compare the expression levels of Pontin between cardiomyocytes and cardiac myoblasts, the authors showed stronger expression of Pontin in cardiac myoblasts than in cardiomyocytes. Furthermore, cardiac myoblasts were found to express Pontin in both the nucleus and cytosol, unlike cardiomyocytes which express Pontin only in the cytosol (Fig 1G). This suggests that Pontin might function in cardiac myoblasts, potentially playing a role in maintaining cardiac homeostasis. Therefore, it is essential to exclude possible effects of knockout of Pontin in cardiac fibroblasts.

We agree with the reviewer that it is important to assess Pontin expression in cardiac fibroblasts in our Pontin^{icKO} mice. To address this issue, we analysed Pontin expression in cardiac fibroblasts isolated from adult Pontin^{icKO} and WT mice as controls. As shown in figure 4B of the revised manuscript, we found that Pontin expression in cardiac fibroblasts of Pontin^{icKO} mice was not different compared to WT controls. This data indicates that the phenotypes of Pontin^{icKO} were likely due to Pontin depletion in cardiomyocytes.

We have added new data (Fig.4B) and corresponding text in the revised manuscript (line 182-183, Results, and line 562-572, Methods).

3. According to Fig. 1A, the expression of Pontin is high in neonatal hearts, but it decreases in adult hearts. However, many experiments were conducted by overexpressing Pontin in neonatal rat cardiomyocytes (NRCM), where Pontin was expected to be highly expressed, or by inducing knockout of Pontin in adult mice, where Pontin was expected to be expressed at a lower level. The authors need to check whether overexpression of Pontin in adult rat cardiomyocytes leads to more pronounced phenotypes.

We follow the reviewer's recommendation and have conducted experiments on adult cardiomyocytes. We isolated adult rat cardiomyocytes (ARCM) and overexpressed Pontin in these cells. We have added new data in Fig.8A-E on the effects of Pontin overexpression in adult rat cardiomyocytes (ARCM). We found that Pontin overexpression in ARCM resulted in significant increases in YAP activity (~11 folds), cell proliferation as indicated by Ki-67 staining (~4.25 folds), and YAP nuclear translocation by ~3.3 folds. The increase in YAP activity and cell proliferation due to Pontin overexpression was more pronounced in ARCM than in NRCM.

We have added new figures and corresponding text to the manuscript in relation to this comment as follows:

- *Addition of new figures 8A-E.*
- *Addition of text in the manuscript in line 327-333 (Results), line 468-470 (Discussion), line 545-561 (Methods).*

4. The authors utilized rat, human, and Rhesus monkey samples in Fig1. However, for the mechanistic study, the authors exclusively used a rat cell line. It raises the question of whether the mechanism by which Pontin expression induces YAP activity is applicable to other species. The authors should utilize Pontin conditional TG mice

samples for immunoblot analysis to measure p-MOB1 levels and for immunohistochemistry (IHC) to evaluate YAP, p-YAP, etc.

We thank the reviewer for the suggestion. To address this comment, we have added new data showing the levels of YAP and MOB1 phosphorylation in Pontin^{cTG} mice. As shown in the revised Fig. S12C-D, we observed significant reductions in pMOB1/MOB1 and pYAP/YAP ratios in Pontin^{cTG} mice compared to WT. This data is consistent with our findings from experiments using cardiomyocytes shown in Figs. 7I and 7M, which showed reduced YAP and MOB1 phosphorylation in neonatal rat cardiomyocytes following Pontin overexpression.

We have added corresponding text in the manuscript to explain the additional data (line 355-361).

5. A mouse model overexpressing Flag-tagged Pontin specifically in cardiomyocytes was generated to demonstrate the protective function of Pontin against AngII-induced pathological cardiac remodeling. It would have been beneficial to utilize this mouse model for further in vivo investigation of Pontin's function. Given their observation of reduced protein expression of Pontin under cardiac stress in Figs. 1B, 1C, and 1H, it would be more reasonable to present the phenotype (Fig. 8) and the altered Hippo pathway to elucidate how Pontin protects against pathological remodeling through Hippo pathway modulation. Additionally, performing RNA-seq and validating the altered Hippo pathway in the Pontin overexpressing mouse model could complement the results obtained from the icKO mouse model.

We agree that it is important to understand the detailed mechanism(s) of Pontin-mediated protection in cardiomyocytes following Ang-II stimulation. We have added new data showing that MOB1 and YAP phosphorylation were reduced in Pontin^{cTG} mice (please see response to comment no. 5 above), indicating regulation of the Hippo pathway in Pontin^{cTG} mice. In addition, as suggested by the reviewer, we performed RNAseq experiments to analyse the transcriptomic profiles of Pontin^{cTG} and WT littermates following Ang-II treatment for 2 weeks. The RNAseq analysis is presented in the new figure 10. The data indicated that the Hippo pathway was the top ranked signalling pathway that was differently regulated in Pontin^{cTG} mice (GSEA analysis, Fig. 10C-D); this might result in the activation of TEAD transcription factor (IPA analysis, Fig. 10E), which is one of the main downstream effectors of the Hippo/YAP pathway. The IPA analysis also showed differential expression of gene clusters related to specific cellular functions, as described in Fig. 10B. Overall, the findings suggested a modulation of the hippo pathway in Pontin^{cTG} mice, which is consistent with the analysis of Pontin^{icKO} mice.

We have modified the manuscript as follows:

- *Addition of text line 376-391 (Results) and line 728-730 (Methods)*
- *Addition of new figure 10.*

6. While Pontin expression was observed to decrease in pathological conditions across various mammalian heart tissues (Fig. 1B-D), it is important to present an appropriate marker that represents each pathological condition. Additionally, it is

recommended to include a marker for phenylephrine treatment in NRCM (Fig. 1H) to ensure proper treatment.

We thank the reviewer for the feedback and would like to provide more information regarding the samples used in Fig. 1B-D. The mouse heart tissue samples were from the TAC-induced cardiac hypertrophy model, a model that was routinely conducted in our lab and resulted in a ~1.5 folds increase in the heart weight/tibia length ratio, as indicated in our previous publication¹. The Rhesus monkey samples were obtained from Dr. Rui-Ping Xiao of Peking University². The pathological phenotypes of these Rhesus monkeys were described in detail in³, which shows phenotypes of hypertension, hyperglycemia, and obesity. The human heart failure samples were obtained from Asterand. Information on the clinical conditions of the donors is included in the revised supplementary table S1. The phenylephrine-induced cardiomyocyte hypertrophy model is routinely conducted in our lab and induces cell enlargement by ~1.3 folds, as indicated in our previous publication¹.

We have included additional text (lines 96-97, 99-100, 103-104, 110-111) to clarify the pathological conditions of the samples presented in figures 1B-D. We also added a new supplementary table S1 describing the clinical conditions of human heart failure samples.

7. Following the RNA-seq data acquired from Pontin icKO mice, the authors confirmed the upregulation of specific Hippo target genes through qRT-PCR analysis. To provide additional confirmation of Hippo pathway activation, it would be beneficial for the authors to perform active YAP staining on mouse heart tissues. This complementary approach would strengthen the evidence for Hippo pathway activation and enhance the comprehensiveness of their study.

We thank the reviewer for raising this important point. To address this issue, we conducted immunohistochemistry analysis to examine the expression of phospho-YAP and total-YAP in cardiac tissue sections of Pontin^{icKO} and control mice. The data presented in Fig.6K-L of the revised manuscript showed a trend of increased level of pYAP/YAP ratio although it did not reach statistical significance. Increased pYAP/YAP level indicated an inhibition of YAP activity, which was consistent with the data that Pontin^{icKO} displayed higher activation of the Hippo pathway that resulted in the inhibition of YAP activity. The lack of statistical significance might be due to the high variability within the Pontin^{icKO} mice. This might be caused by the variation of the level of Pontin gene deletion between mice in this group.

We have modified the manuscript to address this comment as follows:

- *Addition of new figures 6K-L*
- *Addition of text, line 290-294 (Results), line 715-725 (Methods)*

8. It is important to elucidate the underlying molecular mechanism that connects p-MOB1 and Pontin, whether they have a direct relationship or if there is an intermediary mediator bridging their interaction. Further investigation is required to uncover the intricacies of this molecular pathway and to establish the causal relationship between these components.

To establish the link between Pontin and MOB1, we utilised a mutant Pontin construct lacking ATPase activity. It was shown previously that the ATPase activity of Pontin is important for its function, and it can be abolished by replacing the aspartic acid residue at position 302 with asparagine (D302N)⁴. We then compared the level of MOB1 phosphorylation between NRCM overexpressing wild-type (AdPontin) and mutant Pontin (AdPontinD302N). As expected, we observed a significant reduction of MOB1 phosphorylation in NRCM-overexpressing Pontin. In contrast, the MOB1 phosphorylation level in NRCM overexpressing mutant Pontin was not different from controls (Fig.8F-G of the revised manuscript). These suggest that Pontin ATPase activity is essential for Pontin-mediated MOB1 regulation and indicate a direct link between Pontin activity and MOB1 phosphorylation.

We have added this data to Fig.8F-G and included corresponding text in the revised manuscript to explain this finding (line 334-338, Results, line 459-460, Discussion, and line 580-585, Methods).

9. In Figures 7M and S7B, the authors observed a decrease in the phosphorylation level of MOB1 in Pontin overexpressing cells, but no discernible difference in the expression of YAP target genes. To bolster their findings, the authors should undertake further validation of the molecular mechanisms underpinning this observation with additional experiments or analyses.

Thank you for raising this important point. To strengthen our finding that Pontin modulates pathological cardiac remodelling through regulation of the Hippo pathway, we have performed additional experiments, some of which were also suggested by other reviewers. We showed that Pontin lacking ATPase activity did not reduce MOB1 phosphorylation (see also response to comment no. 8 above). In addition, considering that we have shown that lack of Pontin expression caused upregulation of MST1 and MOB1 (Fig 6G and J) and downregulation of active YAP (Fig 7J and K), we therefore investigated whether treatments of NRCM with either MST1 inhibitor XMU-MP1, siRNA to knockdown MOB1, or YAP activator TT-10 can rescue phenotypes caused by Pontin knockdown. As shown in Fig.8H-J XMU-MP1, siMOB1 and TT-10 were capable in rescuing the reduction of YAP activity due to Pontin knockdown. This data indicates that Pontin modulates several components of the Hippo pathway (e.g., by increasing expression and/or activation of MST1 and MOB1), which overall result in YAP activation.

We have added new figures 8H-J and included additional text in the result (line 339-345) to explain this finding.

10. To strengthen their findings in Figures 2 and 3, the authors should conduct rescue experiments by reintroducing Pontin into the knockdown cell lines. This would help further elucidate the role of Pontin and provide additional insights into the mechanisms in their study.

We have followed the reviewer's recommendation and compared the level of YAP activity in NRCM lacking Pontin and following Pontin reintroduction using recombinant adenovirus. As shown in the revised Fig.S11A, YAP activity increased after re-expression of Pontin using adenovirus, suggesting that the observed changes in YAP activity can be rescued by Pontin overexpression.

We have added figure S11A and added new text (line 302-303) to include this finding in the revised manuscript.

Minor points

1. It is crucial to confirm whether the same image was utilized for both MOB1 and YAP immunoblot data in Figure 6F.

Thank you for pointing this out. We apologise for this mistake. We have replaced the MOB1 immunoblot data accordingly (Fig.6F).

2. It is essential to present the immunofluorescence data in a manner that includes not only the merged view but also individual channels displayed separately.

We now present images of the split channels from immunofluorescence analysis. See figures 1H, 1J, S1, S2, S4 in the revised manuscript.

3. In Figure 7C (upper) and Figure 7E (upper), both images serve as control references, and therefore, they should exhibit similar characteristics. However, it is noteworthy that in Figure 7E (upper), the localization of YAP within the nucleus appears to be less distinct when compared to Figure 7C (upper).

We understand the reviewer's concern; however, it is important to note that the GFP-YAP data shown in figures 7C (upper) and 7E (upper) were from different experiments, even though both served as controls. Figure 7C (upper) was from NRCM treated with adenovirus expressing LacZ, whereas Fig 7E (upper) was from NRCM transfected with scrambled siRNA. Thus, the difference in the GFP localization might be due to the difference in the treatment above.

Reviewer #2 (Remarks to the Author):

The authors show that endogenous Pontin expression in cardiomyocytes plays a protective effect in the heart and that the protective effect of Pontin may be mediated through activation of YAP.

General:

Although the authors suggest that Hippo inactivation may mediate the detrimental effect of Pontin downregulation in cardiomyocytes in the mouse model, the results are only correlative, and the involvement of other mechanisms cannot be excluded.

Although in vitro experiments with neonatal myocytes may show that Pontin has cell-autonomous effects in cardiomyocytes, the quality of the data is generally modest.

We thank the reviewer for the feedback. To address these comments, we have performed a number of experiments, such as rescue experiments using Hippo inhibitors and YAP activator, experiments using inactive mutant of Pontin, and

additional RNAseq analysis of Pontin transgenic mice. All of the new data from these experiments strengthened the notion that Pontin exerts its function in the cardiomyocytes via modulation of the Hippo/YAP pathway. Details of the new experiments and the findings are explained in the responses to specific comments from reviewers 1, 2, and 3.

However, we agree that the involvement of other pathways cannot be totally excluded. Further studies need to be done to fully characterise the regulatory role of Pontin in cardiomyocytes. We have added new sentences in the discussion to acknowledge this matter. (See line 486-492 in the revised manuscript).

Specific:

1. In Figure 1, the authors should isolate cardiomyocytes and non-myocytes from the mouse heart and show the level of pontin in cardiomyocytes and non-myocytes separately. The data shown in Fig. 1G is unclear and does not allow the comparison of the protein levels among various cell types.

To address this comment, we have included additional data (western blot analysis) showing the expression of Pontin in neonatal cardiomyocytes and cardiofibroblasts. As can be seen in the revised figure 1G Pontin expression in cardiofibroblasts was higher than in cardiomyocytes.

We also modified figures 1H and 1J to include images from separate channels to improve the clarity of Pontin subcellular localization in cardiomyocytes and cardiofibroblasts.

2. In Figures 2 and 3, the quality of the experiments is modest. The quality of the cell staining is modest and there are many non-myocytes in the culture. The Y axis in some graphs is not presented in a full scale in Fig. 2. Thus, the effect of pontin modulation is modest.

*We thank the reviewer for the feedback. To address this comment, we included images from individual (split) channels (this was also suggested by reviewer 1) in supplementary figure S1, S2 and S4, so that the green signal can be seen more clearly. We improved the signal exposure in the RGB merge image to show that most of the cells were positive for α -actinin (cardiomyocyte marker). We modified graphs in figures 2 and 3 so that all Y axes are at full scale. However, it is important to note that all of the effects of Pontin overexpression or knockdown were statistically significant. The effect size in vitro might not be enormous; however, we believe that might be enough to alter the phenotype as suggested in the in vivo observations of *Pontin^{icKO}* and *Pontin^{cTG}* mice.*

3. In Figure 4, the authors should show cell type-specific downregulation of Pontin in cKO mice.

This issue was also raised by reviewer 1. Please see the response to comment no. 2 of reviewer 1.

4. In Figures 5E and 6, the authors should show the level of phosphorylation of Mst1 and YAP in cardiomyocytes.

We have examined the level of phosphorylated MST1 and YAP in cultured cardiomyocytes with Pontin overexpression and gene silencing. As shown in figures 7G, 7J, 7M, and 7N MST1 phosphorylation was not affected by Pontin gene overexpression or silencing; however YAP phosphorylation was significantly altered by Pontin overexpression and gene silencing. We agree that it is important to examine if the change in YAP phosphorylation level also occur in vivo in Pontin^{icKO} and Pontin^{cTG}. Immunohistochemistry analysis as described in figure 6K-L showed a trend of increased phospho-YAP level in Pontin^{icKO}, whereas immunoblot shown in figure S12C indicated a decrease in phospho-YAP level in Pontin^{cTG} mice.

5. Although the study suggests that cardiomyocyte-specific downregulation of Pontin may stimulate the Hippo pathway. How it affects the activity of YAP in the cKO heart is not clearly shown. Again, the activity of YAP should be assessed in a cardiomyocyte-specific manner.

This issue was also raised by reviewer 1. Please see comment no.7 of reviewer 1 and our response to address it.

6. In Figure 8, the authors should show the level of pontin overexpression in cardiomyocytes in PontincTG. The immunoblot analyses should be conducted with anti-Pontin antibody, not anti-Flag. It would also be nice if the authors could show to what extent Pontin overexpression in PontincTG mice rescues the level of Pontin expression in cardiomyocyte during stress.

We have followed the reviewer's suggestion. We have included Western blot data showing Pontin detection using Pontin antibody in Pontin^{cTG} mice, both in basal condition and after stimulation with Ang II. We included these new data as Figure S12A-B. We also modified the text in the revised manuscript line 354-355.

Minor:

The authors should have addressed as to how pontin downregulation leads to the activation of the Hippo pathway and inhibition of YAP.

We have shown in the manuscript that cardiomyocyte specific ablation of Pontin increases expressions of the core components of the Hippo pathway, including MST1 and MOB1. In addition, in vitro experiments using NRCM suggested that Pontin knockdown induces MOB1 phosphorylation. Together, both mechanisms may contribute to the activation of the Hippo pathway and the inhibition of YAP.

We clarified this issue in the revision (lines 334-345).

Reviewer #3 (Remarks to the Author):

The manuscript submitted by Dr. Oceandy and colleagues presents interesting observations related to the Pontin gene in adult cardiac physiology. In order to study the role of Pontin in the mouse heart, the authors generated both an inducible cardiac-specific knock-out and a cardiomyocyte-specific overexpression of Pontin. Knock-out of Pontin generated hypertrophic phenotypes and exacerbated Angiotensin-II treatment, while overexpression had minimal baseline effects, but ameliorated Angiotensin-II treatment. Between cell culture systems and the in vivo models, the authors identify Pontin may be regulating core Hippo signaling components (both abundance and post-translational modifications) with Pontin levels positively correlating with YAP activity.

Specific comments:

1. In Figure 4, it is described that the phenotype emerges between the 3rd and 4th week after tamoxifen. What happens after 4 weeks post-tamoxifen? Do the animals continue to get worse or does the phenotype stabilize at ~40% EF? It seems like 4 weeks was a premature endpoint.

We thank the reviewer for raising this important point. The majority of Pontin^{icKO} mice died due to cardiac failure after 4 weeks post-tamoxifen injection. Therefore, we analysed the cardiac phenotype and terminated the experiment at 4 weeks post-tamoxifen injection.

2. In the text related to the RNA-seq (line 210), it states that the cut-off was P-value < 0.05 and log2 fold change >1 or <-1, but in Figures 5A and 5B, the Y-axis and dotted line appear to be in relation to the false discovery rate. Please confirm the method of cut-off, and if it was a P-value cut-off, please justify not using the multiple testing correction.

We used FDR values for the cut-off. We apologise for the confusion. We have revised the manuscript accordingly (line 245).

3. In Figure 5, is there any connection between the genes observed at 1 week post-tamoxifen and the genes observed at 3 weeks post-tamoxifen? Was the ~400 differentially expressed genes nested within the ~2000 differentially expressed genes?

To address this comment, we analysed differentially regulated genes (DEGs) in Pontin^{icKO}, comparing 1 week vs. 3 weeks post-tamoxifen injection. We found 60 genes that were differentially regulated at both 1 week and 3 weeks after tamoxifen injection. To describe the functions of these genes, we performed functional enrichment analysis based on gene ontology database. The sets of genes that are most differently expressed at both 1 and 3 weeks after tamoxifen are genes related to the regulation of the cell cycle and cell division. This data is consistent with the in vitro and in vivo findings showing that Pontin regulates cell proliferation via the Hippo pathway.

We have added figure S10A-B in the revised manuscript and added text in line 247-251 to address this comment.

4. In Figures 6-7, the authors are suggesting that Pontin acts upstream of Hippo signaling to regulate YAP activity; however, there are some oddities that are making it unclear about the relationships between core members of the Hippo cascade. For example, there are increased levels of total MST1 and total MOB-1 and phosphor-MOB-1, but not downstream components like LATS1/2. Using the NRCMs, can the authors further flesh out the interaction between Pontin, Hippo signaling pathway, and YAP? Specifically, can Pontin knockdown phenotypes (Figures 2&3) be rescued by MOB1 knockdown? Can Pontin knockdown phenotypes be rescued by LATS1/LATS2 knockdown? Can Pontin knockdown be rescued with constitutively active YAP? Can the effects of Pontin overexpression be reversed by YAP knockdown (is this direct YAP activity)?

We have conducted additional experiments to address this comment. We analysed whether a reduction in YAP activity due to Pontin knockdown can be rescued by MOB1 knockdown. As shown in figure 8I we observed significantly increased YAP activity in Pontin-deficient cells following MOB1 knockdown. We then tested if activating YAP using the YAP activator TT-10 and inhibiting MST1 using XMU-MP-1 could also rescue the phenotype. The data presented in figures 8H and 8J confirmed the increase in YAP activity following YAP activation or MST1 inhibition. Together, our additional data strengthen the idea that Pontin acts as a modulator of the Hippo/YAP pathway.

We have added figures 8H-J and text in line 339-345 in the revised manuscript to clarify this issue.

REFERENCES

1. Zi M, *et al.* The mammalian Ste20-like kinase 2 (Mst2) modulates stress-induced cardiac hypertrophy. *J Biol Chem* **289**, 24275-24288 (2014).
2. Song R, *et al.* Central role of E3 ubiquitin ligase MG53 in insulin resistance and metabolic disorders. *Nature* **494**, 375-379 (2013).
3. Zhang X, *et al.* Rhesus macaques develop metabolic syndrome with reversible vascular dysfunction responsive to pioglitazone. *Circulation* **124**, 77-86 (2011).
4. Zhao Y, *et al.* Pontin, a new mutant p53-binding protein, promotes gain-of-function of mutant p53. *Cell Death Differ* **22**, 1824-1836 (2015).

REVIEWER COMMENTS

Reviewer #1 (Remarks to the Author):

The authors properly addressed my concerns by providing newly designed experimental data or reasonable explanation by citing previous literature. I would like to request that the authors include the following significant papers on the function of Pontin in the reference section of the revised manuscript.

1. Yu YS, Shin HR, Kim D, Baek SA, Choi SA, Ahn H, Shamim A, Kim J, Kim IS, Kim KK, Won KJ, Baek SH. Pontin arginine methylation by CARM1 is crucial for epigenetic regulation of autophagy. *Nat Commun.* 2020 Dec 8;11(1):6297. doi: 10.1038/s41467-020-20080-9.

2. Tarangelo A, Lo N, Teng R, Kim E, Le L, Watson D, Furth EE, Raman P, Ehmer U, Viatour P. Recruitment of Pontin/Reptin by E2f1 amplifies E2f transcriptional response during cancer progression. *Nat Commun.* 2015 Dec 7;6:10028. doi: 10.1038/ncomms10028.

3. Boo K, Bhin J, Jeon Y, Kim J, Shin HJ, Park JE, Kim K, Kim CR, Jang H, Kim IH, Kim VN, Hwang D, Lee H, Baek SH. Pontin functions as an essential coactivator for Oct4-dependent lincRNA expression in mouse embryonic stem cells. *Nat Commun.* 2015 Apr 10;6:6810. doi: 10.1038/ncomms7810.

4. Rottbauer W, Saurin AJ, Lickert H, Shen X, Burns CG, Wo ZG, Kemler R, Kingston R, Wu C, Fishman M. Reptin and pontin antagonistically regulate heart growth in zebrafish embryos. *Cell.* 2002 Nov 27;111(5):661-72. doi: 10.1016/s0092-8674(02)01112-1.

The authors adequately addressed my concerns, and the revised manuscript becomes suitable for publication in *Nature Communications*.

Reviewer #2 (Remarks to the Author):

The authors addressed many issues raised by the reviewers and the paper is improved.

Since the main conclusion and the novelty of this study is the role of pontin in mediating YAP, the authors could have shown that the effect of cardiomyocyte pontin downregulation is mediated by YAP by conducting YAP rescue experiments. Can the downregulation of YAP mechanistically mediate the effect of pontin downregulation? The current study shows only correlative evidence in vivo.

The authors could have provided a more thorough analysis of endogenous pontin in vivo models. It would be important to show how either loss or gain of pontin function in their mouse models affects the level of endogenous pontin. The level of pontin in cardiomyocytes needs to be presented in both control and loss/gain of function mice at baseline and in the presence of Ang II cardiomyopathy. If pontin is overcompensated in the Ang II model in vivo, it may not be physiological.

The authors should show how the level and the activity of YAP was affected in vivo in the presence or absence of pontin overexpression in vivo in the presence or absence of Ang II.

The authors could have strengthened discussion as to how pontin affects the upstream component of the Hippo signaling. What is the connection between TEAD and the Hippo upstream components.

Reviewer #3 (Remarks to the Author):

the authors have done a large amount of experiments to address my concerns and the concerns of the other reviewers. Overall this is a much improved manuscript that has a compelling story

RESPONSES TO REVIEWERS' COMMENTS

Reviewer #1:

The authors properly addressed my concerns by providing newly designed experimental data or reasonable explanation by citing previous literature. I would like to request that the authors include the following significant papers on the function of Pontin in the reference section of the revised manuscript.

1. Yu YS, Shin HR, Kim D, Baek SA, Choi SA, Ahn H, Shamim A, Kim J, Kim IS, Kim KK, Won KJ, Baek SH. Pontin arginine methylation by CARM1 is crucial for epigenetic regulation of autophagy. *Nat Commun.* 2020 Dec 8;11(1):6297. doi: 10.1038/s41467-020-20080-9.
2. Tarangelo A, Lo N, Teng R, Kim E, Le L, Watson D, Furth EE, Raman P, Ehmer U, Viatour P. Recruitment of Pontin/Reptin by E2f1 amplifies E2f transcriptional response during cancer progression. *Nat Commun.* 2015 Dec 7;6:10028. doi: 10.1038/ncomms10028.
3. Boo K, Bhin J, Jeon Y, Kim J, Shin HJ, Park JE, Kim K, Kim CR, Jang H, Kim IH, Kim VN, Hwang D, Lee H, Baek SH. Pontin functions as an essential coactivator for Oct4-dependent lincRNA expression in mouse embryonic stem cells. *Nat Commun.* 2015 Apr 10;6:6810. doi: 10.1038/ncomms7810.
4. Rottbauer W, Saurin AJ, Lickert H, Shen X, Burns CG, Wo ZG, Kemler R, Kingston R, Wu C, Fishman M. Reptin and pontin antagonistically regulate heart growth in zebrafish embryos. *Cell.* 2002 Nov 27;111(5):661-72. doi: 10.1016/s0092-8674(02)01112-1.

The authors adequately addressed my concerns, and the revised manuscript becomes suitable for publication in Nature Communications.

Thank you very much for the very positive comments. We agree that the above papers are very important in the understanding of Pontin in regulating key cellular functions. In the revised manuscript, we have cited and included these papers in the references:

- *Yu et al. is now cited as reference 36. The text reads: "In other cell types, Pontin regulates cell cycle/mitotic progression^{33, 34}, chromatin remodelling and transcription regulation³⁵, nuclear regulation of autophagy through epigenetic mechanism³⁶, DNA damage response³⁷, and maintaining ES cell pluripotency³⁸, indicating its involvement in various biological processes." (lines 411-415).*
- *Tarangelo et al. is cited as reference 35, in these texts:*
 - *Lines 411-415 (please see above)*
 - *Lines 416-419: " Pontin was identified as a key modulator of the cell cycle in glioma³⁹, was involved in E2f-mediated cancer progression in a model of hepatocellular carcinoma³⁵, and has been suggested as a potential diagnostic and prognostic marker of oral squamous cell carcinoma⁴⁰ and diffuse large B cell lymphoma⁴¹.*
- *Boo et al. is now included as citation 38. Please see lines 411-415 as above.*
- *Rottbauer et al. has been included as reference 14 in this manuscript and is cited in various parts of the manuscript, such as Introduction (lines 72-79), Results (lines 122), and Discussions (lines 400, 483).*

Reviewer #2:

The authors addressed many issues raised by the reviewers and the paper is improved.

We thank Reviewer 2 for acknowledging that our paper is improved.

Since the main conclusion and the novelty of this study is the role of pontin in mediating YAP, the authors could have shown that the effect of cardiomyocyte pontin downregulation is mediated by YAP by conducting YAP rescue experiments. Can the downregulation of YAP mechanistically mediate the effect of pontin downregulation? The current study shows only correlative evidence in vivo.

We thank the reviewer for the suggestion. To further support the idea that Pontin regulates YAP, we performed rescue experiments by using an inducible-expression system to express active YAP (YAP^{S127A}) in cardiomyocytes that are deficient of Pontin (siRNA-mediated). As shown in Fig.7M, induced overexpression of YAP^{S127A} reversed YAP activity due to Pontin knockdown. YAP^{S127A} overexpression also rescued the reduction of cell proliferation due to Pontin knockdown, as indicated by the level of mitosis marker Ki-67 expression (Figs.S11B-C). Additionally, reduced YAP activity due to Pontin knockdown can also be rescued by administration of specific YAP activator TT-10 (Fig.8J). Together, this data strongly indicated that the cardiomyocyte phenotypes observed following Pontin knockdown are mainly mediated by YAP.

We have added new data in the revised manuscript (Figs.7M; S11B-C)) and edited the corresponding text in the results section (lines 274-282).

The authors could have provided a more thorough analysis of endogenous pontin in vivo models. It would be important to show how either loss or gain of pontin function in their mouse models affects the level of endogenous pontin. The level of pontin in cardiomyocytes needs to be presented in both control and loss/gain of function mice at baseline and in the presence of Ang II cardiomyopathy. If pontin is overcompensated in the Ang II model in vivo, it may not be physiological.

To address this comment, we analysed Pontin levels in the hearts of wild type (WT) and Pontin^{cTG} mice separately. Our data in Fig.S13A shows a strong trend of Pontin reduction in WT mice in response to Ang-II treatment. This finding is in line with the data presented in Figs 1B-D&I, showing that Pontin expression is significantly reduced in several different pathological conditions, including pressure-overload induced hypertrophy in mice, metabolic syndrome in non-human primates, human heart failure, and isolated cardiomyocytes treated with phenylephrine. On the other hand, expression of Pontin in Pontin^{cTG} mice did not significantly change following Ang-II treatment (Fig. S13B). It is possible that Pontin transgene expression, which is driven by the α MHC promoter, compensates for the reduction of endogenous Pontin expression following Ang-II treatment, and this may provide protective effects to the heart in pathological conditions.

We have added these new data in the revised manuscript (Figs.S13A-B) and added corresponding text in the results section (lines 340-345).

The authors should show how the level and the activity of YAP was affected in vivo in the presence or absence of pontin overexpression in vivo in the presence or absence of Ang II.

We thank the reviewer for the feedback. To address this issue, we have added new data on YAP activation (phospho/total-YAP) in WT and Pontin^{cTG} mice following treatment with Ang II or saline (as a control)(Fig.S13C). In the control condition (saline treatment), the level of phosphorylated YAP was reduced in Pontin^{cTG} mice compared to WT, indicating an increase in YAP activity. p-YAP level was reduced in WT-Ang-II compared to the WT-saline group (indicating increased activity in WT

mice following Ang-II treatment). However, p-YAP level did not significantly change in *Pontin^{cTG}*-Ang-II compared to the *Pontin^{cTG}* saline group. These data suggested that YAP was already activated in *Pontin^{cTG}* mice before stimulation with AngII. This might explain the protective effects against pathological stimuli in these mice since YAP has a strong anti-apoptotic effect. Treatment with Ang-II increased YAP activity in WT mice, presumably as an adaptive response against pathological stress; however, this treatment did not alter p-YAP level in *Pontin^{cTG}* mice, possibly because YAP activity was already high at basal condition in *Pontin^{cTG}* mice.

We modified figure S13C and the manuscript text accordingly (line 345-348).

The authors could have strengthened discussion as to how pontin affects the upstream component of the Hippo signaling. What is the connection between TEAD and the Hippo upstream components.

Our data showed that Pontin regulates core components of the Hippo pathway, in particular MOB1. Expression of Pontin reduced MOB1 phosphorylation, while Pontin gene silencing increased MOB1 phosphorylation. Interestingly, expression of inactive mutant Pontin^{D320N} did not change MOB1 phosphorylation. Together, these results support the idea that Pontin regulates the upstream component of the Hippo pathway. The discussion around these findings has been included in lines 440-445.

*TEAD is the main transcription factor that is activated by YAP (Ma S, et al. Annu Rev Biochem. 2019; 88:577-604). Since YAP activation is tightly regulated by Hippo core components (MST1/2, LATS1/2, MOB1, and SAV1), TEAD activity is also regulated by Hippo upstream components. That might explain the RNASeq data that TEAD is differentially regulated in *Pontin^{cTG}* mice with the highest significance (Fig. 10E).*

Reviewer #3:

The authors have done a large amount of experiments to address my concerns and the concerns of the other reviewers. Overall, this is a much improved manuscript that has a compelling story.

We thank Reviewer 3 for the positive feedback.

REVIEWERS' COMMENTS

Reviewer #2 (Remarks to the Author):

No further comments